

# Early warning and drought risk assessment for the Bolivian Altiplano agriculture using high resolution satellite imagery data

Claudia Canedo Rosso[1,2], Stefan Hochrainer-Stigler[3], Georg Pflug[3,4], Bruno Condori[5], and Ronny Berndtsson[1,6]

[1]Division of Water Resources Engineering, Lund University, P.O. Box 118, SE-22100 Lund, Sweden
[2] Institute of Hydraulics and Hydrology, Mayor University of San Andres, Cotacota 30, La Paz, Bolivia
[3] International Institute for Applied Systems Analysis (IIASA), Schlossplatz 1, A-2361 Laxenburg, Austria
[4] Institute of Statistics and Operations Research, Faculty of Economics, University of Vienna, Oskar-Morgenstern-Platz 1, 1090 Wien, Austria
[5] National Institute of Agriculture and Forestry Innovation (INIAF), Batallón Colorados 24, La Paz, Bolivia.
[6] Center for Middle Eastern Studies, Lund University, P.O. Box 201, SE-22100 Lund, Sweden.

*Correspondence to*: Claudia Canedo Rosso (canedo.clau@gmail.com and claudia.canedo_rosso@tvrl.lth.se)

**Abstract.** Implementation of agriculturally related early warning systems is fundamental for the management of droughts. Additionally, risk-based approaches are superior in tackling future drought hazards. Due to data-scarcity in many regions, high resolution satellite imagery data are becoming widely used. Focusing on ENSO warm and cold phases, we employ a risk-based approach for drought assessment in the Bolivian Altiplano using satellite imagery data and application of an early warning system. We use a newly established high resolution satellite dataset and test its accuracy as well as performance to similar (but with less resolution) datasets available for the Bolivian Altiplano. It is shown that during the El Niño years (warm ENSO phase), the result is great difference in risk and crop yield. Furthermore, the Normalized Difference Vegetation Index (NDVI) can be used to target specific hot spots on a very local scale. As a consequence, ENSO early warning forecasts as well as possible magnitudes of crop deficits could be established by the government, including an identification of possible hotspots during the growing season. Our approach therefore should not only help in determining the magnitude of assistance needed for farmers on the local scale but also enable a pro-active approach to disaster risk management against droughts that can include economic-related instruments such as insurance as well as risk reduction instruments such as irrigation.

Keywords: El Nino, ENSO, Satellite Imagery Data, Drought risk, Quinoa,Potato

## 1 Introduction

Agricultural production is highly sensitive to weather extremes, including droughts and heat waves. Losses due to such extreme hazard events pose a significant challenge to farmers as well as governments worldwide (UNISDR, 2015). Worryingly, the scientific community predicts an amplification of these negative impacts due to future climate change (IPCC, 2013). Especially in developing countries, drought can be seen as a major natural hazard. However, the impacts vary on a seasonal and annual timescale, on the hazard intensity, and the capacity to prevent and respond to droughts (UNISDR, 2009). Regarding the former,



the El Niño Southern Oscillation (ENSO) plays an important role. For example, between 1964 and 1993 half of the global population were affected by disasters due to droughts and impacts associated with El Niño Southern Oscillation (ENSO) occurrences (Bouma et al., 1997). Also more recently, in 2015-2016, the strong warm phase ENSO (El Niño) triggered droughts in several regions around the world, driving losses in agricultural crops and increased food insecurity (Kogan and

Guo, 2017). Regarding the prevention aspect, there are several strategies and policies possible to reduce socioeconomic and environmental consequences of drought events (WMO, 2006). These include the setup of insurance and irrigation systems. Facing increasing agricultural losses at all scales, from the local to the global level, whatever strategies will be implemented in the future, the governments have to take a leading role (Lal et al., 2012). Indeed, the recent Sendai Framework for Disaster Risk Reduction (SFDRR) agreement was developed to significantly reduce global disaster mortality, number of affected

people, economic loss, and damage to critical infrastructure and disruption of basic services due to natural disasters (including droughts) by 2030. The agreement explicitly recognizes the need for governments to have a primary role in reducing disaster risk with support of relevant stakeholders (UN, 2015). The SFDRR explicitly calls for an increase in the availability and access to disaster early warning systems. In fact, drought early warning systems are essential components for drought preparedness and resilience because they can provide information to water-sensitive sectors such as farming in a timely manner. The

Sustainable Development Goals (SDGs) state that the priority areas for adaptation to climate change are water and agriculture. These in turn, are related to the largest climate hazards that are floods, drought and higher temperatures (UN, 2016). Additionally, the implementation of early warning is fundamental for drought disaster risk management, proactive planning, and mitigation policy measures in vulnerable regions, including Latin American countries such as Bolivia (Verbist et al., 2016). Based on these observations, the objective of our paper is to present ways forward for a risk-based early warning system to

mitigate drought events using observed gauge data and high resolution satellite based imagery data for the Bolivian Altiplano. As in most cases in developing countries, data scarcity is a major issue for data monitoring, tracking progress, and the evaluation of risk reduction strategies (major goals within the SFDRR and SDGs, (see UN, 2015, 2016). Our study is no exception and we provide a first assessment and baseline estimates that should be embedded within an iterative risk management process so that they can be improved when new data become available. Such an approach is now seen as the most

appropriate in data-scarce environments (see IPCC, 2012; and UN, 2015). The South American Altiplano is special as it is besides Tibet the largest high plateau in the world, located in the central part of the Andes mountain chain. Based on the last national census in Bolivia in 2012, the Bolivian Altiplano has about 2,000,000 inhabitants (INE, 2015b). Most important rainfed crops in the region include quinoa and potato. Climate variability in the Altiplano is strongly associated with the El Niño Southern Oscillation (ENSO), and droughts are generally driven by the ENSO

warm phases (Vicente-Serrano et al., 2015; Garreaud and Aceituno, 2001; Thompson et al., 1984). Generally, agricultural productivity in the Altiplano is low due to high susceptibility to the climate, poor soil conditions, and the mainly manual labour. Poor agricultural production is associated with the ENSO climate phenomena (Buxton et al., 2013). Apart from that, high evaporative demand and erratic distribution of rainfall increase agricultural vulnerability significantly. In addition,





precipitation dependency for the rainfed agriculture, and frost risk limit farming production to the moist austral summer (Garcia et al., 2007;Condori et al., 2014).

Large socio-economic losses have been experienced in the past in Bolivia due to ENSO. For example, 135,000 people were affected and USD 515 million loss were registered in 1997-1998 (UNDP, 2011). More recently, 665,000 people were affected
and over USD 450 million of losses in 2015-2016 were experienced (Guha-Sapir et al., 2016). To lessen the long term impacts of these events, the national government has allocated large budget amounts for emergency operations to compensate part of the losses, which are usually evaluated in an ex-post fashion. Important in our context is that Bolivia implemented a national program for risk disaster management that includes an early warning system that informs about disaster occurrence a few days ahead the event. However, in case of a drought event, the warning time is not enough to prevent losses in the agricultural
sector. Based on ENSO forecasting, an El Nino event can be predicted 1 to 7 months ahead (Tippett et al., 2012). For this time period it may be possible to implement ex-ante policies to reduce societal vulnerability to droughts, stressing preparedness, and improve risk management strategies. We are especially interested in how a risk based approach can be used to determine the potential need of resources as well as ways to determine hotspots where it is likely that these resources need to be distributed. A constraint to study drought occurrence as well transform ex-post strategies into proactive activities is the uneven
and scarce distribution of weather and crop related ground data in the region. We therefore used new precipitation and vegetation satellite data that present full coverage of the spatial distribution in the study area. We combine this new information with ground data to enhance the knowledge and provide consistent results for climate and vegetation variability. The new dataset is tested for its suitability by comparing with previously used datasets in the regions. We show how forecasting risk based crop production within an early warning system can be used to mitigate impacts for spatially explicit regions.

Our paper is organized as follows. Given the importance of data limitations in our case study region, we first introduce in section 2 the recently available datasets that we employed for our analysis. In section 3 we discuss the methods employed to (i) test the validity of the new datasets to be used in our analysis, (ii) show how the relationship between climate and vegetation data was estimated, and (iii) how ENSO was incorporated in our analysis. After this, section 4 presents results and our proposed framework. Finally, section 5 ends with a conclusion, ways forward, and outlook to the future.

**2. Case Study Region and Data Availability**

The Bolivian Altiplano covers about 150,000 km$^2$. It contains more than 70% of the total Altiplano surface area, the remaining percentage is located in southern Peru and northern Chile. La Paz, Oruro, and Potosi are the major administrative regions in the Bolivian Altiplano. The major city of the region is El Alto, with about 850,000 inhabitants, followed by Oruro with 265,000 inhabitants. The Altiplano has a pronounced south-north gradient (200–900 mm year$^{-1}$) in annual precipitation with the wet
season occurring from November to March (Garreaud et al., 2003). Over 60% of the annual precipitation occur during the summer months (DJF) in association with the South American Monsoon (SAM) (see Fig. A1). However, inter-annual variations are large (Garreaud, 1999;Lenters and Cook, 1999). The rainfall scarcity is more severe in the southern Altiplano,



where the precipitation is almost zero during non-summer months, in contrast the northern Altiplano presents not only heavier but also more evenly distributed rainfall over the year (Fig. A1).

In the Altiplano, important indigenous civilizations emerged including the Inca and Tihuanaco. These civilizations domesticated plants and animals such as potato, quinoa, and lama (Erickson, 1988;Repo-Carrasco-Valencia and Serna, 2011).

Today, agriculture is still the main economic activity in this region. However, crop production is limited to the summer due to water constraints and climate conditions (Garcia et al., 2007). Nowadays, less than 20% of the agricultural area are irrigated in the Bolivian Altiplano (INE, 2015a). Therefore, surface water is the main water source for agricultural production. Common agricultural production in the area is focused on crops, fishing, and cattle (UNESCO-WWAP, 2003). Quinoa and potato are two of the most important crops. As already indicated above, large socio-economic losses are generated due to drought events

as well as the impact of ENSO in Bolivia but also in other Latin American countries (Federman et al., 2014;Aguilar-Barajas et al., 2016). Due to the fact that farming is mostly rainfed, weather related phenomena are very important for crop production and can serve as a good predictor for risk. As will discussed next, data availability on the local scale is very limited. However, new recent datasets essentially needed for our approach have been made available as discussed below.

## 2.1 Climate Variables: Precipitation and Temperature Data

Time series of monthly precipitation at 23 locations and mean, maximum, and minimum temperature at 15 locations from July 1981 to June 2016 were obtained from the National Service of Meteorology and Hydrology (SENAMHI; see Appendix Table A1). Initially, the available precipitation data sets included 65 gauges but only 23 had less than 10% of missing data. Some small data gaps where filled with the mean monthly value of the whole dataset to provide full time series. Outliers were identified by comparing with neighbouring monthly values. The inter-annual temperature at the 15 locations varied

considerably between summer (DJFM) and winter (JJAS), including a larger variance for the minimum temperature (Fig. A2a). The mean monthly maximum temperature is about 15°C. Mean monthly minimum temperature is lower than 5°C, and in winter monthly temperature decreases to -17°C in some regions. Regions close to the Lake Titicaca present lower inter-annual variability; this is also the case for Copacabana [10] (Fig. A2b). In Copacabana, the mean monthly maximum temperature oscillates from 12 to 15°C during the year, and the corresponding mean monthly minimum temperature is about 1 to 5°C. In

this region the temperature is normally above 0°C. In contrast, Uyuni (22) close to Uyuni Salt Flat presents larger oscillations (Fig. A2c). In this region, mean maximum temperature during summer months is above 20°C and during winter months about 12 to 14°C. The Uyuni mean monthly minimum temperature differences between summer and winter months are even larger. In summer, they oscillate between 0 to 6°C and in winter between -7 to -12°C.

It should be noted that the precipitation gauges have an uneven spatial distribution and are mainly concentrated in the northern

Bolivian Altiplano. To improve the spatial coverage of rainfall data, monthly quasi-rainfall time series from satellite data were included in our study. The Climate Hazards Group InfraRed Precipitation with station data (CHIRPS) quasi-global rainfall dataset was used. CHIRPS presents a 0.05° resolution satellite imagery and is a quasi-global rainfall dataset from 1981 to the near present with a satellite resolution of 0.05° (Funk et al., 2015). CHIRPS dataset builds on interpolation techniques and





high resolution, long period of record precipitation estimates based on infrared Cold Cloud Duration (CCD) observations. CCD values are a measure of the amount of time a given pixel has been covered by high cold clouds. CHIRPS uses the Tropical Rainfall Measuring Mission Multi-Satellite Precipitation Analysis (TMPA 3B42) 2000–2013 with 0.25° of resolution to calibrate global Cold Cloud Duration (CCD) rainfall estimates.

## 2.2 El Niño Southern Oscillation Data

The Oceanic Niño Index (ONI) is usually used to identify El Niño (warm) and La Niña (cool) years (http://www.cpc.ncep.noaa.gov/). ONI is the 3 month running mean of Extended Reconstructed Sea Surface Temperature (ERSST v5) anomalies in the El Niño 3.4 region. The El Niño 3.4 anomalies represent the average equatorial SSTs in the equatorial Pacific Ocean (5ºN to 5ºS latitude, and 120º to 170ºW longitude). Five consecutive overlapping three month periods at or above +0.5°C anomaly represents warm events (El Niño), and at or below the -0.5 anomaly are cold (La Niña) events.

## 2.3 Crop Production and Vegetation Data

As indicated, quinoa and potato are the main crops in the Bolivian Altiplano and still gaining importance. The Quinoa growing season is from September to April, for potato it is from October to March. Data of quinoa and potato yield were obtained from the National Institute of Statistics (INE) in Bolivia from July 1981 to June 2016 for La Paz, Oruro, and Potosi (Fig. 1). No crop yield data on the local scale are available and this is a major limitation that needs to be addressed in the future. The annual datasets represent production (t) in relation of the area (ha) at regional level.

Additionally, the Normalized Difference Vegetation Index (NDVI) was used in our study in order to relate climate, vegetation and crop production. NDVI can estimate the vegetation vigour (Ji and Peters, 2003) and crop phenology (Beck et al., 2006). Considering the coarse resolution of the regional crop data, the use of vegetation data could improve the spatial resolution of crop information during different phenological phases. This is possible as vegetation data are available at bimonthly time steps (i.e., every 15 days) with a spatial resolution of 0.08˚. NDVI was assembled from the Advanced Very High Resolution Radiometer (AVHRR) sensors by the Global Inventory Monitoring and Modelling System (GIMMS). As a result, the NDVI 3g.v1 (third generation GIMMS NDVI from AVHRR sensors) data set was defined. This bimonthly data set spans from July 1981 to December 2015. Note, the NDVI is an index that presents a range of values from 0 to 1, bare soil values are closer to 0, while dense vegetation has values close to 1 (Holben, 1986). However, NDVI datasets have snow, cloud, and interpolation errors. NDVI 3g.v1 GIMMS provides information to differentiate valid values from these possible errors; errors are represented by values larger than 1. For this reason, the values larger than 1 were eliminated within our dataset. The mean monthly values were calculated based on the bimonthly NDVI time step (every 15 days). The 3% of data gaps that remained were filled using the nearest neighbour NDVI values.





## 3. Methodology

The satellite data was firstly compared to the ground-based data. The satellite rain data was tested with gauged precipitation using the same methodology employed (for comparison reasons) in Blacutt et al. (2015) and Satgé et al. (2016) who did a similar analysis for Boliva but with different satellite imagery products. We discuss in the next sections our methodology employed to empirically combine the climate, vegetation, and crop data to derive risk based crop distributions including ENSO effects.

### 3.1 Validation of Satellite Rainfall Product using Gauge-Measured Precipitation Data

The performance of the satellite products in relation to accurately estimating the amount of rainfall was based on statistical measures including categorical statistics to assess rain detection capabilities. For the quantitative statistics, mean error (ME) or bias, and root mean squared error (RMSE) were calculated based on Wilks (2006), and Nash Sutcliffe efficiency (NSE) coefficient based on Nash and Sutcliffe (1970) was used as well. Additionally and similar to Blacutt et al. (2015) and Satgé et al. (2016), we used the Spearman rank correlation to estimate the goodness of fit to observations, the bias to show the degree of over- or underestimation (Duan et al., 2015), the root mean square error (RMSE) to compute the average magnitude of the estimated errors (values closer to zero generally indicate smaller magnitude of error), and finally the Nash Sutcliffe Efficiency coefficient that evaluates the prediction accuracy compared to observations (1 corresponds to a perfect match between gauge observation and satellite-based estimate and zero indicates that the satellite estimations are as accurate as the mean of the observed data; negative values indicate that the observed mean is better than satellite-based estimate, see Nash and Sutcliffe (1970) for more details). To evaluate results, correlation coefficients larger or equal to 0.7 with a significance level of 0.01 were considered as reliable (Satgé et al., 2016;Condom et al., 2011).

Two statistical indicators based on a contingency table were computed for the categorical statistics, namely POD and FAR. Probability of Detection (POD) indicates what fraction of the observed events were correctly estimated, and False Alarm Ratio (FAR) indicates what fraction of the predicted events did not actually occur (Bartholmes et al., 2009;Ochoa et al., 2014;Satgé et al., 2016). The categorical statistic measures were used to evaluate the satellite estimations. Here, the rainfall amounts are considered as discrete values, i.e., rain occurrence or absence of rain. Based on this approach, four scenarios were taken into account: the number of events when the satellite rain estimation and the rain gauge reports a rain event (H), when only the satellite rain estimation reports a rain event but is a false alarm (F), and when only the rain gauge reports a rain event but the satellite not and therefore is a miss (M).

### 3.2 Linear Regression of Vegetation and Climate Variables: Precipitation and Temperature

Multivariate statistical models and methods were applied in order to find the relation between vegetation and climate dimensions. For instance, stepwise regression was used in a combination of forward and backward selection. The independent variable considered in the study was vegetation, and the dependent variables were accumulated precipitation and Accumulated



Degree Days (ADD). Firstly, the NDVI was related to observed and satellite-derived precipitation data. Secondly, the ADD was included in the analysis (Eq. 1).

$$NDVI = \beta_0 + \beta_1 cummulated\ precipitation + \beta_2 accumulated\ degree\ days \qquad (1)$$

Stepwise linear regression was applied to the NVDI as independent variable, and precipitation and temperature as dependent

variables. Both precipitation and temperature were represented as accumulated values (for temperature using the GDD). The mean monthly temperature was multiplied by the number of days of each month to obtain daily values. GDD was computed only considering the months of the growing season for each year. To calculate the ADD, the accumulated value of the Growing Degree Day (GDD) multiplied by the number of days of each month was computed. The GDD is defined as the difference of mean and base temperature (Eq. 2).

$$GDD = \frac{Tmax + Tmin}{2} - T_b \qquad (2)$$

where $T$max and $T$min are monthly maximum and minimum temperature, respectively, and $T_b$ is the minimum threshold or base temperature. Base temperature of potato was 4°C and 3°C for quinoa (Jacobsen and Bach, 1998). If $T_b$ is greater than $T$mean, GDD is equal to 0.

For forward selection, the variables were entered into the model one at a time in an order determined by the strength of their

correlation with the criterion variable. The effect of adding each variable was assessed as it was entered, and variables that did not significantly add to the fit of the model were excluded (Kutner et al., 2004). For backward selection, all predictor variables were entered into the model. The weakest predictor variable was then removed and the regression re-calculated. If this significantly weakened the model then the predictor variable was re-entered, otherwise it was deleted. This procedure was repeated until only useful predictor variables (in a statistical sense) remained in the model (Rencher, 1995). The results were

compared with other results from the literature to check for suitability of results with phenology and weather related dimensions of plants.

### 3.3 Relationship of Vegetation and Crop Yield

In the Bolivian Altiplano, quinoa and potato annual yields from 1981 to 2015 are available for three regions: La Paz, Oruro, and Potosi. We suggest that the coarse distribution of the crop yield data can be improved using the NDVI. Besides improving

the crop yield resolution, the NDVI also allowed to analyse the variability of vegetation at a monthly time scale. The NDVI dataset was available at a bimonthly time scale from July 1981 to June 2016. To define the surface area for agricultural production in the Bolivian Altiplano, a land use map of the region was used (Fig. 1). The Bolivian Altiplano is composed of the Titicaca, Desaguadero, Poopó, and Salar de Coipasa System (TDPS), and the Uyuni Basin. The land use map for the TDPS was developed by the Autonomous Authority of the Lake Titicaca in 1995 at a scale of 1:250,000 (UNEP, 1996). The land use

map for Uyuni Basin was developed by the Ministry of Development Planning in 2002 (geo.gob.bo) using Landsat imagery and ground information at a scale 1:1,000,000. Figure 1 shows the land use map for the Altiplano. The bare land considers: exposed rock, sand, bad land, and fluvial deposits. The wetlands are: hydromorphic soil, marshes, and bofedales. Additional





grazing land are: pasture and shrub; and surface water includes: lake and rivers. The crop land, including potato and quinoa crops, represents about 12% of the total surface area.

## 3.4 Relationship between ENSO and Crop Yield

The Oceanic Niño Index (ONI) was used to identify El Niño (warm) and La Niña (cool) years
(http://www.cpc.ncep.noaa.gov/). The threshold was further broken down into weak (with a 0.5 to 0.9 SST anomaly), moderate (1.0 to 1.4), and strong (≥ 1.5) events (http://ggweather.com/enso/oni.htm). In this study we considered the categories neutral/moderate (with a 0 to 1.4 SST), strong El Nino (≥ 1.5) and strong La Nina (≤ -1.5) years (Appendix Table A2). The classification considered three consecutive overlapping 3-month periods at or above the +1.5ºC anomaly for warm (El Niño) events and at or below the -1.5ºC anomaly for cold (La Niña) events. The ENSO year considered in this study starts in June-
July-August and ends in May-June-July for each year from 1981 to 2016. Subsequently, the crop yield of quinoa and potato was compared with strong El Nino years. This relationship was analyzed using two sample t-test and Wilcoxon rank sum test. Two sample t-test and Wilcoxon rank sum test compare two independent data samples, with the difference that the first compare samples that have normal distribution, and the second one is a non-parametric test (Wilks, 2006). Here, the null hypothesis of the two sample t-test is that the crop yield during El Nino and neutral/moderate years have equal means. And
the null hypothesis of the Wilcoxon rank sum test is the crop yield during El Nino and neutral/moderate years are samples from continuous distributions with equal medians. Both tests computes two-sided p-value. When the hypothesis is equal to 1, the null hypothesis is rejected at 5% significance level. And the null hypothesis is accepted when it is equal to zero.

## 4. Results and discussion

### 4.1 Validation of Satellite Rain Data using Gauge-Measured Precipitation Data

The advantage of using CHIRPS is the higher spatial resolution of the data, i.e., CHIRPS presents a resolution of 0.05°, obtained with the resampling of TMPA 3B42 (with 0.25° grid cell). This spatial resolution represents a better option for agricultural studies. Validation of the satellite rain data using empirical precipitation data from the weather stations (Fig. 1) was developed for the 23 locations where gauge precipitation data are available (Table 1). The mean error (ME) calculation showed that most of the stations present a bias of +/-10 mm/month, meaning that the satellite simulations generally under- or
overestimate 10 mm in relation with the ground precipitation data, however Charazani location presented a large bias of 40.7 mm/month. The RMSE in 80% of the locations had a range between 15 to 30 mm/month, Charazani [6] had a RMSE of 58 mm/month. NSE analyses showed that most of the stations presented values from 0.9 to 1, however Ayo Ayo [3], Charazani [6], and Collana [8] presented low NSE values (-0.6, 0.6, and 0.7). The spearman rank correlation between ground observed precipitation data and satellite rain product was significant (P<0.01) for all locations. Most of the correlation coefficients were
higher than 0.7. The locations Charazani [6] and Colcha K [7] presented a significant (P<0.01) correlation of 0.65.



The quantitative methods discussed in section 3.1 were applied at monthly time step for the 23 gauge locations. The relative ME (Fig. A3a) results were +/-1 [-], except in April-June. This means that CHIRPS simulations under- or overestimate by less than 10% compared to observed precipitation. The relative RMSE shows the average magnitude of the estimated errors (Fig. A3b), CHIRPS has a relative RMSE close to zero from December to March, thus it adequately simulated the rainfall, especially

during this period when about 70% of the total annual precipitation occur. This is corroborated by, e.g., (Garreaud et al., 2003). The relative RMSE for July and June present large values, meaning that the satellite rain data do not simulate properly the precipitation during these months, and the precipitation is frequently sub estimated. However, the precipitation often is very low in July and June (with an average close to zero, see Fig. A1), and therefore the amount of precipitation during these months has low significance.

Seasonal precipitation for summer (DJF), autumn (MAM), winter (JJA) and spring (SON) showed that relative ME is +/-0.5 [-] and relative RMSE is lower than 1 during summer, autumn, and spring. Good fit is evident for summer. As mentioned before this season is the most important because of the cropping season. Summarizing, the results showed that satellite rain from CHIRPS and observed precipitation represent acceptable values for most of the studied sites and performed at least as good as other datasets currently available. As a consequence, we used this satellite product for our further studies.

**4.2 Linear Regression: NDVI and Climate Variables**

The precipitation in the Altiplano shows a marked rainy season from November to March. The highest peak of precipitation is in January for the 23 observed precipitation locations (Fig. 2a). NDVI displays the highest peak in March and April (Fig. 2b). The lag between the max precipitation and max NDVI is reasonable since vegetation requires time to grow (e.g. Shinoda, 1995;Cui and Shi, 2010;Chuai et al., 2013). The cumulative precipitation was calculated for a period of 12 months from July

to June of the following year for all locations. The spearman rank correlation coefficient for accumulated precipitation and NDVI with a lag of two months was from 0.63 to 0.79 with statistical significance at 0.001 level. The Tihuanacu [21] location presented the highest correlation (Fig. A4). The stations presented a linear relation between precipitation and NDVI, until NDVI reached its highest value (1). Hence, for this value, precipitation has no longer effect on vegetation.

The results for stepwise linear regression between NDVI and accumulated precipitation were statistically significant at the

0.01 level. The analysis was firstly applied with observed precipitation and afterwards with satellite data. The correlation coefficient oscillated from 0.4 to 0.8 in both cases, and most of the stations presented coefficients higher than 0.6. Additionally, stepwise linear regression for NDVI as independent variable, and the accumulated precipitation and GDD as dependant variables was performed. GDD was computed using the difference of mean and base temperature of the crop. Figure A5 shows the average of mean monthly temperature for the 15 locations. It was calculated using the arithmetic difference between

maximum and minimum monthly temperature. The results of the stepwise regression using Eq. (2) show statistical significance for all 15 locations included in the study. The correlation coefficients oscillated from 0.5 to 0.8 and therefore were further used in our study.



### 4.3 Relationship between Vegetation and Crop Yield

In La Paz, a total of 85 NDVI locations are situated on the surface crop land area of the land use map, Oruro represents 106 locations, and Potosi 15 locations. The mean of the maximum annual NDVI datasets were computed for La Paz, Oruro, and Potosi, and compared with the annual quinoa and potato yields from 1981 to 2015 (Fig. A6). Spearman rank correlation was used to find the relationship between the mean of the maximum NDVI and crop yield (Table 2). As a result, La Paz and Potosi showed significant correlation between NDVI and crop yield, higher correlation was found between NDVI and potato yield. Oruro presents no significant correlation (P>0.05).

### 4.4 Relationship between ENSO and Crop Yield

The relationship between ENSO and crop yield was analysed using two sample t-test and Wilcoxon Rank-Sum Test. To test the relationship, crop yield during El Nino years (warm ENSO phase) was compared with crop yield during neutral/moderate years (see Table A2). The results show that quinoa yield during warm ENSO phase and neutral/moderate years present a significant difference at a 95% confidence level (Table A3). The difference of quinoa yield (t/ha) between El Nino and neutral/moderate years is about -0.2 t/ha. This means that the production during neutral/moderate years is higher. Quinoa yield during El Nino and neutral/moderate years presents no significant difference at Oruro. Part of this result could be explained by the employing of more advanced crop management strategies (e.g., selected crop varieties and the application of agricultural innovations) as this region is one of the largest producer in Bolivia and the world (Ormachea and Ramirez, 2013). The t-test and Rank-Sum test results show that potato yield during El Nino and neutral/moderate years are significantly different at 95% confidence level at Potosi (Table A3). In Oruro this relation is significant for the t-test, and for the rank sum test it is significant at the 92% of confidence level. The difference in potato yield (t/ha) between El Nino and neutral/moderate is about -0.5 and -0.9 t/ha for Oruro and Potosi, respectively. The results show that production during neutral/moderate years is also higher.

A kernel density estimation procedure was used to define the probability density function for the crop production during El Nino and neutral/moderate years (Fig. 3). Quinoa yield in La Paz and Potosi presents a peak in the density function at ~0.6 t/ha during neutral/moderate years, and during the El Nino years two peaks are shown at ~0.45 and ~0.2 t/ha. The last peak represents the yield during very strong El Nino periods during 1982-1983, 1991-1992 and 1997-1998. As mentioned above, these periods had a large impact on agricultural losses in the region. As well, potato yield density function for El Nino and neutral/moderate years shows differences. The peak for the production in La Paz is at ~6 t/ha, in Oruro it is ~4 t/ha, and Potosi it is 5 t/ha during neutral/moderate years. However, during El Nino years the peak reduces to ~5.7 t/ha, ~3.5 t/ha, and ~4.5 t/ha in La Paz, Oruro, and Potosi, respectively. The second peak representing crop yield during very strong El Nino years shows half of the normal crop yield.



## 4.5 Discussion: Magnitude of Assistance Needed during Drought Events and Distribution of Aid

We employed a new high spatial resolution satellite dataset and tested it for accuracy as well as performance to similar (but with coarser resolution) datasets available for our region. Using this dataset it was shown that during El Nino years large differences in crop yield and risk (Fig. 3) exist. Furthermore, it was found that NDVI can be related to crop yield and therefore

NDVI could be used to target specific hot spot depending on NDVIs available on a local scale. As a consequence, ENSO early warning forecasts as well as possible magnitudes of crop deficits could be established by the government, including identification of possible hotspots during the growing season. Our approach therefore, should not only help for determining the magnitude of assistance needed for farmers on the local level but also enable a pro-active approach to disaster risk management against droughts. This may include not only economic related instruments such as insurance but also risk

reduction instruments such as irrigation needs. In fact, early warning based financing is gaining increasing attraction in some real world settings as it has several advantages. However, it should be acknowledged that large challenges exists (French and Mechler, 2017).

A major aim of a drought early warning system is to detect the probability of occurrence and severity of a drought. Correspondingly, it should assist to reduce impacts through mitigation and preparedness measurements including the active

participation of decision makers, e.g., through the implementation of policies and programmes (WMO, 2006). Drought severity could be measured via shifts from normal conditions of climatic parameters such as precipitation. As was done in our case, we not only provided shifts but the difference in risk for El Nino and neutral/moderate years. However, one of the main challenges of drought early warning systems is data-scarcity, e.g., low density or not evenly distributed stations for hydro-meteorological data networks, poor data quality due to missing data, and restricted use of data between government agencies or other

institutions. Although, a drought event is caused by the lack of precipitation, there are numerous indicators that determine its intensity and duration. Therefore, drought evaluation should combine multiple climate, water, and soil parameters and socio-economic indicators to define its potential impact. As it was shown here, ENSO warm phase related characteristics are especially important in the context of extreme drought events and should therefore be incorporated within early warning systems as standard practice.

Despite these challenges for development of early warning systems, applications have been successful in the past. For example, in Uganda, the NGO ACTED and the national Government implemented a drought early warning system for Karamoja by collecting data and monitoring to predict droughts including a periodically information about drought risks and catalysing the application of preparedness measures (UNISDR, 2015). The early warning system developed by the Beijing Climate Center in China evaluates drought occurrence using the standardized precipitation index, and provides drought-related information to

policy makers and stakeholders for the definition of mitigation measurements (WMO, 2006). There are numerous cases in many countries around the globe. As in our case, particularly in the mid-latitudes weather patterns are strongly influenced by ENSO. Monitoring and predicting ENSO can therefore significantly contribute to reduce the risk of disasters through early warning systems implementation (IPCC, 2012;FAO, 2017).



The availability of risk estimates is especially useful to employ advanced risk management approaches (Pflug and Werner, 2007). An iterative risk management or adaptive management approach that uses monitoring, research, evaluation, learning, and innovation to improve risk management strategies over time seems especially relevant in a data scarce environment (IPCC, 2012). The ongoing assessment, action, reassessment, and response to new data, dynamic crop models and instruments allows

the continuous adaptation to risk and ultimately to a better management of hazards.

## 5. Conclusions

This study is a first attempt to provide an agricultural drought risk assessment in relation with ENSO. Given the large differences in risk, early warning indicators and corresponding strategies to lessen the impacts could be implemented in the Bolivian Altiplano. In doing so, we introduced and tested a new high resolution satellite product which was used for estimating

crop risk. Additionally, a significant influence of precipitation on vegetation and crop yields in the region was identified. The ENSO impact on crop production was evaluated by studying the relation of vegetation and climate variables, considering that El Nino generally drives a drought event. Our study provides valuable information for early warning systems, primarily by providing information of the relationship between crop production and vegetation, and subsequently a relation between vegetation and climatological parameters. Moreover, we showed that ENSO phases are strongly related with crop yield, and

with this information the prediction of ENSO could be used to define risks in terms of decrease in crop yields in the studied region. While overall good fit among climate, ENSO, and crop yield variables were found, it is important to consider other parameters, such as evapotranspiration and soil moisture in improved models. With such information also agricultural models could be set up and risk management plans with better accuracy determined.

**Acknowledgements:**

Part of this research was done during the Young Scientist Summer Program (YSSP) 2017 of the International Institute for Applied System Analysis (IIASA). Part of this research was supported by FORMAS Research Council for Environment, Landscaping and Urban Development, and the Swedish International Development Cooperation Agency (SIDA). The authors would like to express their gratitude the Servicio Nacional de Meteorologia e Hidrologia (SENAMHI) for providing the

meteorological data.

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





## APPENDIX

**Table A1. Spatial location of the 23 weather stations where gauged precipitation data is available, 15 stations also present temperature maximum and minimum data (T).**

| $N_o$ | Station name | Region | Latitude | Longitude | Altitude | T |
|------|--------------|--------|----------|-----------|----------|---|
| [1] | Achiri | La Paz | -17.21 | -69.00 | 3880 | |
| [2] | Ancoraimes | La Paz | -15.90 | -68.90 | 3882 | |
| [3] | Ayo Ayo | La Paz | -17.09 | -68.01 | 3888 | T |
| [4] | Berenguela | La Paz | -17.29 | -69.21 | 4145 | |
| [5] | Calacoto | La Paz | -17.28 | -68.64 | 3830 | T |
| [6] | Charazani | La Paz | -15.20 | -69.00 | 3659 | |
| [7] | Colcha K | Potosí | -20.74 | -67.66 | 3780 | T |
| [8] | Collana | La Paz | -16.90 | -68.28 | 3911 | T |
| [9] | Conchamarca | La Paz | -17.38 | -67.46 | 3965 | |
| [10] | Copacabana | La Paz | -16.17 | -69.09 | 3870 | T |
| [11] | El Alto Aeropuerto | La Paz | -16.51 | -68.20 | 4034 | T |
| [12] | El Belen | La Paz | -16.02 | -68.70 | 3833 | T |
| [13] | Hichucota | La Paz | -16.18 | -68.38 | 4460 | T |
| [14] | Oruro Aeropuerto | Oruro | -17.95 | -67.08 | 3701 | T |
| [15] | Patacamaya | La Paz | -17.24 | -67.92 | 3793 | T |
| [16] | Salla | La Paz | -17.19 | -67.62 | 3500 | |
| [17] | San Jose Alto | La Paz | -17.70 | -67.78 | 3746 | |
| [18] | San Pablo de Lipez | Potosí | -21.68 | -66.61 | 4256 | |
| [19] | Santiago de Huata | La Paz | -16.05 | -68.81 | 3845 | T |
| [20] | Santiago de Machaca | La Paz | -17.07 | -69.20 | 3883 | T |
| [21] | Tiahuanacu | La Paz | -16.57 | -68.68 | 3863 | T |
| [22] | Uyuni | Potosí | -20.47 | -66.83 | 3680 | T |
| [23] | Viacha | La Paz | -16.66 | -68.28 | 3850 | T |

**Table A2. The classification of strong El Nino (≥ 1.5ºC), strong La Nina (≤ -1.5) and neutral/moderate (0 to 1.4) years for the period 1981 to 2016.**

| Strong El Nino | Neutral and moderate | Strong La Nina |
|----------------|----------------------|----------------|
| 1982 | 1981 | 1988 |
| 1986-1987 | 1983-1985 | 1998-1999 |
| 1991 | 1989-1990 | 2007 |
| 1997 | 1992-1996 | 2010 |
| 2015 | 2000-2006 | |
| | 2008-2009 | |
| | 2011-2014 | |
| | 2016 | |



**Table A3. T-test and Wilcoxon rank sum test for quinoa and potato yield during El Nino years and neutral/moderate years. If the hypothesis is equal to 1 it means that we rejected the null hypothesis at a confidence level of 95%.**

|        |        | T test 2 sample | | | Wilcoxon rank sum test | | |
|--------|--------|------------|---------|--------|------------|---------|--------|
|        |        | Hypothesis | P value | t-stat | Hypothesis | P value | z-stat |
| Quinoa | La Paz | 1 | ~0    | 3.38 | 1 | 0.002 | 3.08 |
|        | Oruro  | 0 | 0.31  | 1.02 | 0 | 0.169 | 1.37 |
|        | Potosi | 1 | 0.004 | 3.15 | 1 | 0.010 | 2.56 |
| Potato | La Paz | 0 | 0.080 | 1.80 | 0 | 0.20 | 1.31 |
|        | Oruro  | 1 | 0.003 | 3.23 | 0 | 0.08 | 1.73 |
|        | Potosi | 1 | 0.004 | 3.08 | 1 | 0.02 | 2.31 |

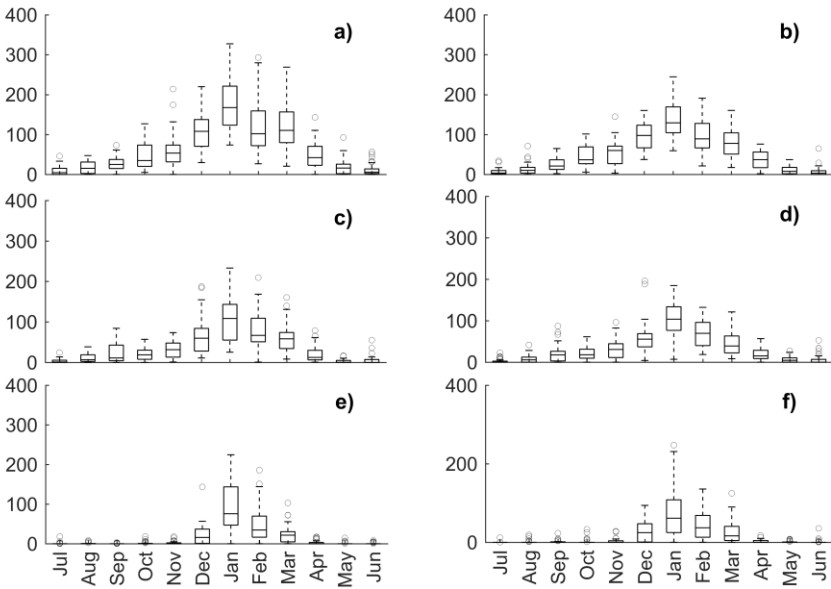

5    **Figure A1. Boxplot of the total monthly precipitation in the Bolivian Altiplano at the northern Altiplano (a) Copacabana station [10], and (b) El Alto station [11], the central Altiplano (c) Oruro station [14], and (d) Patacamaya station [15], and southern Altiplano (e) Colcha K station [7], and Uyuni station [22].**




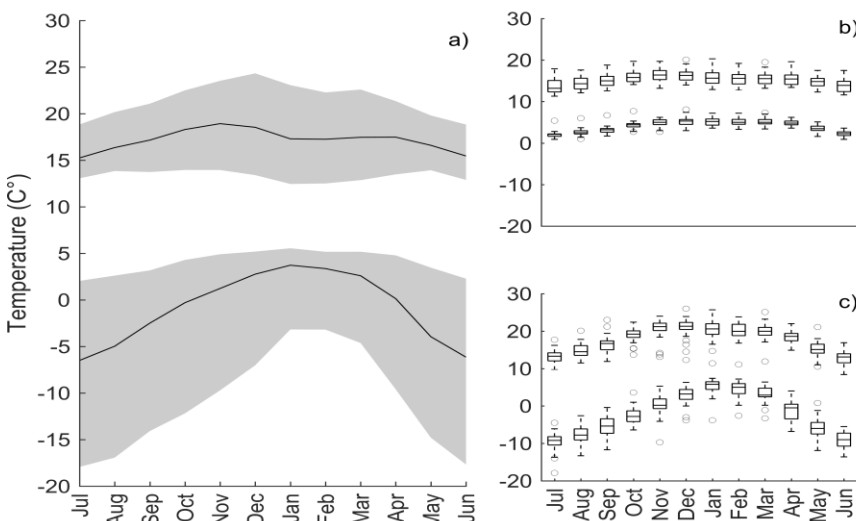

Figure A2. (a) Average of the mean monthly maximum and mean monthly minimum temperature of 15 time series in the Bolivian Altiplano from July 1981 to June 2016 (black solid line). The range of the mean maximum and minimum monthly temperature values from the datasets is shown in shaded area. Boxplot of the monthly maximum and minimum temperature in (b) Copacabana station [10] and (c) Uyuni station [22], respectively.

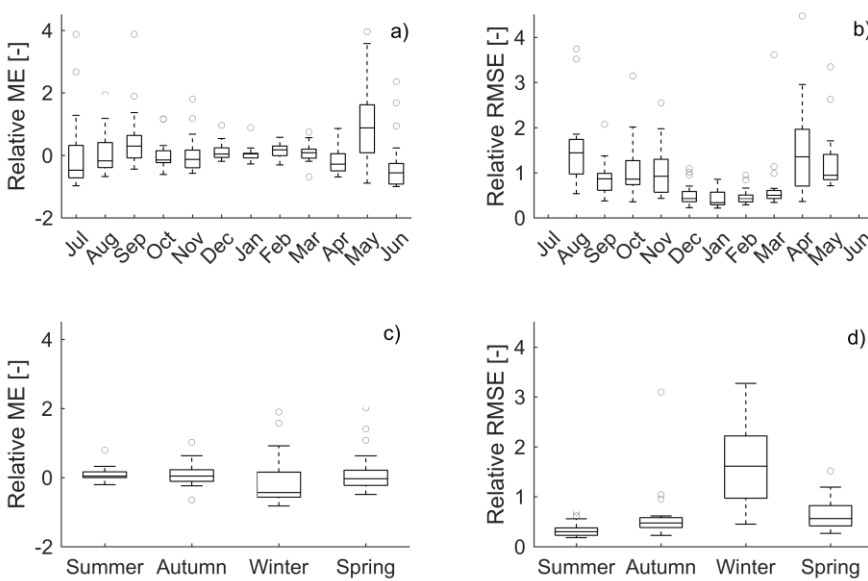

Figure A3. Boxplot of the relative mean error (bias) for the 23 locations at (a) monthly (c) seasonal time scale between satellite rain data and ground observed precipitation data (perfect score 0). And Boxplot of the relative RMSE at (b) monthly and (d) seasonal time scale (perfect score 0). The relative RMSE (b) for July and June present large values in comparison to the other months, for instance the median of the relative RMSE of July and June are 3.2 and 6.8 respectively months, due to the disproportion the boxplots were not included in the figure.



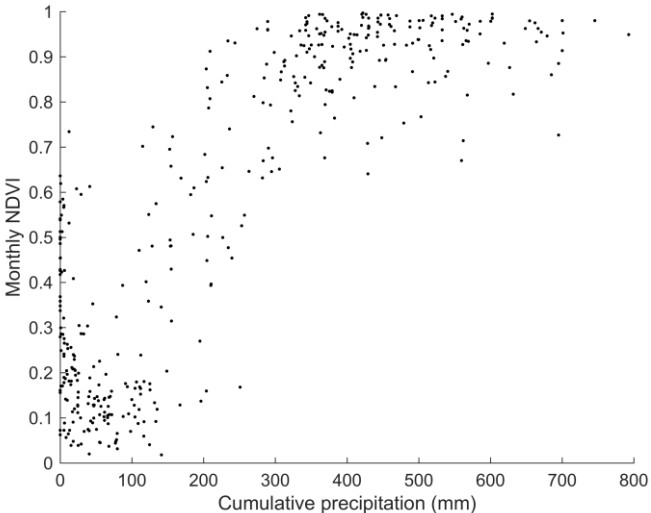

**Figure A4. Scatterplot of the cumulative monthly precipitation and the monthly NDVI with a lag of two months at Tihuanacu station [21].**

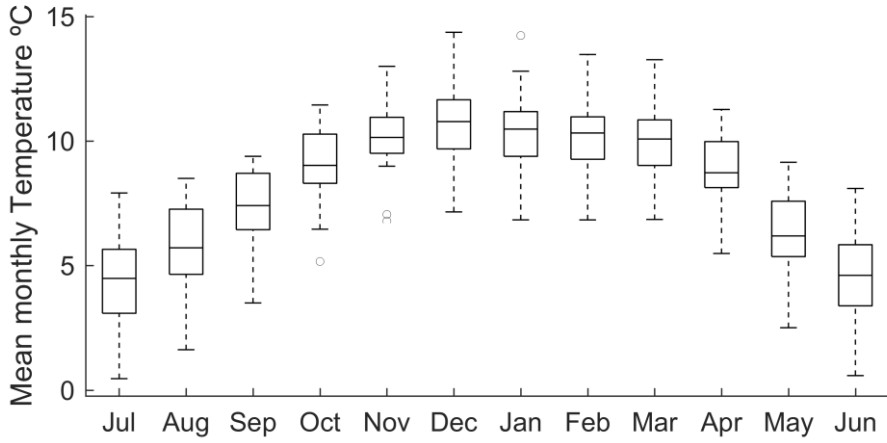

5 **Figure A5. Boxplot of the average of mean monthly temperature at 15 datasets observed at weather stations in the Bolivian Altiplano.**



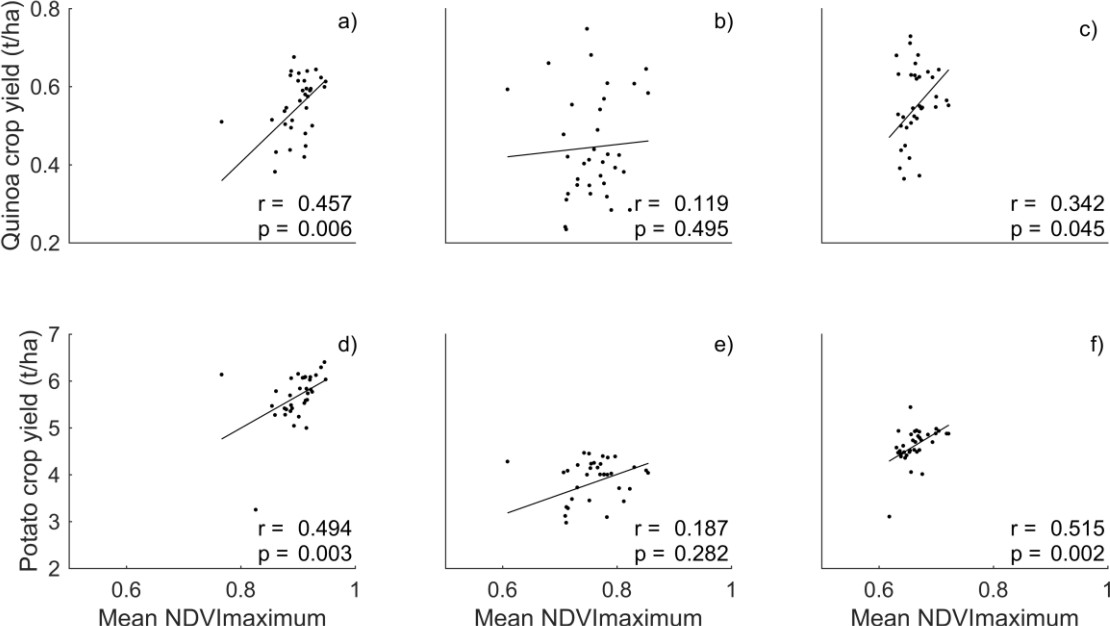

**Figure A6. Scatterplot of the mean annual maximum NDVI and the quinoa yield in La Paz (a), Oruro (b), and Potosi (c). And scatterplot of the mean annual maximum NDVI and potato yield in La Paz (d), Oruro (e), and Potosi (f). Where "r" is the spearman rank correlation coefficients and "p" is the p-value.**





**Table 1. Validation of satellite imaginary rain data using ground observed data of precipitation.**

|  | ME or Bias (mm/month) | RMSE (mm/month) | NSE (-) | Spearman rank correlation | |
|---|---|---|---|---|---|
|  |  |  |  | Coefficient | P value |
| Minimum | -10.5 | 18.0 | -0.6 | 0.65 | <0.01 for all |
| Mean | 2.3 | 28.0 | 0.9 | 0.81 | Locations |
| Maximum | 40.7 | 58.2 | 1.0 | 0.92 | |
| Standard deviation | 10.1 | 10.4 | 0.3 | 0.07 | |
| Perfect score | 0.0 | 0.0 | 1.0 | 1.00 | |

**Table 2. Spearman rank correlation between mean of the maximum annual NDVI and crop yield in La Paz, Oruro, and Potosi.**

|  | Quinoa | | Potato | |
|---|---|---|---|---|
|  | Correlation | P value | Correlation | P value |
| La Paz | 0.46 | 0.006 | 0.49 | 0.003 |
| Oruro | 0.12 | 0.495 | 0.19 | 0.282 |
| Potosi | 0.34 | 0.045 | 0.51 | 0.002 |





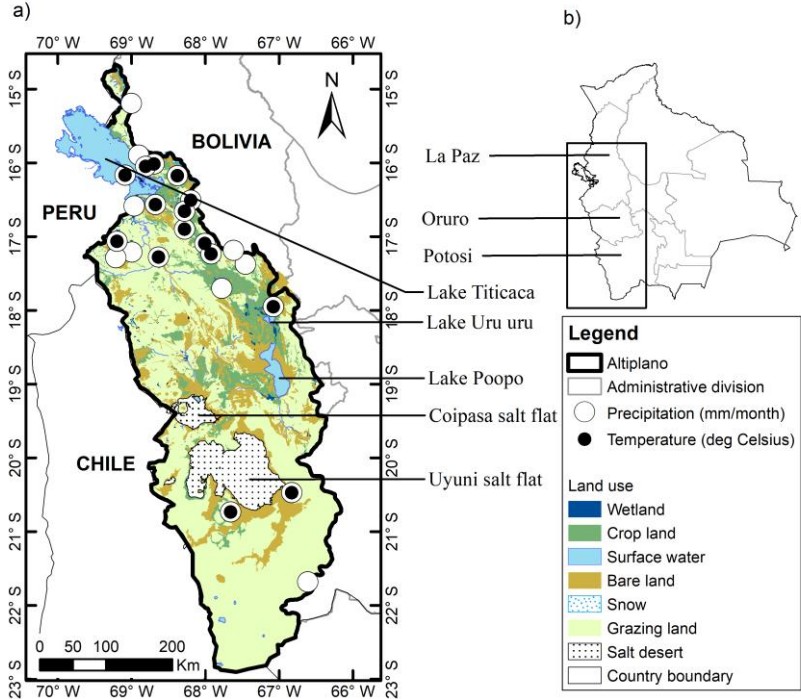

**Figure 1. a) The Bolivian Altiplano (solid black line), and location of weather stations for 23 precipitation gauges (white circle) and 15 temperature gauges (black circle) (b) Major administrative divisions of Bolivia: La Paz, Oruro, and Potosi where crop yield datasets are available.**

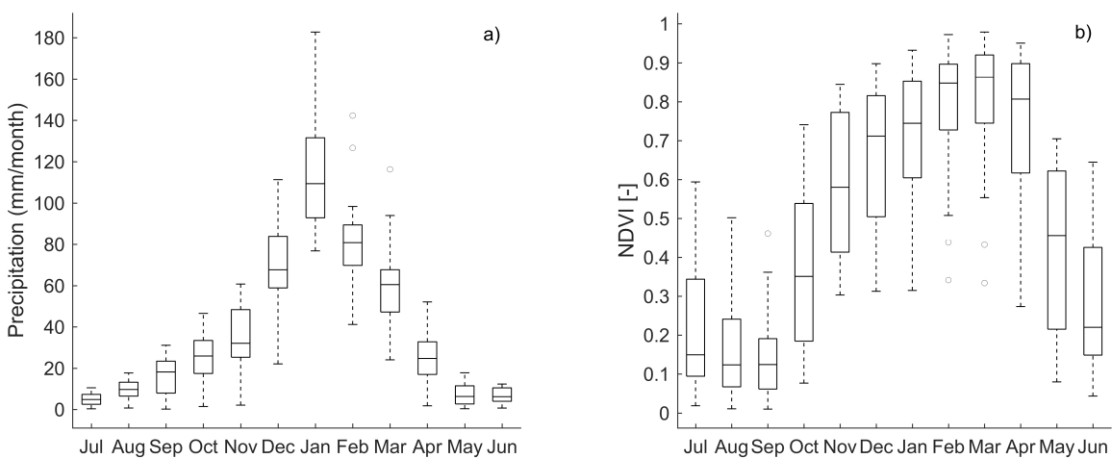

**Figure 2. a) Boxplot of observed mean monthly precipitation and b) mean monthly NDVI at the 23 studied locations.**



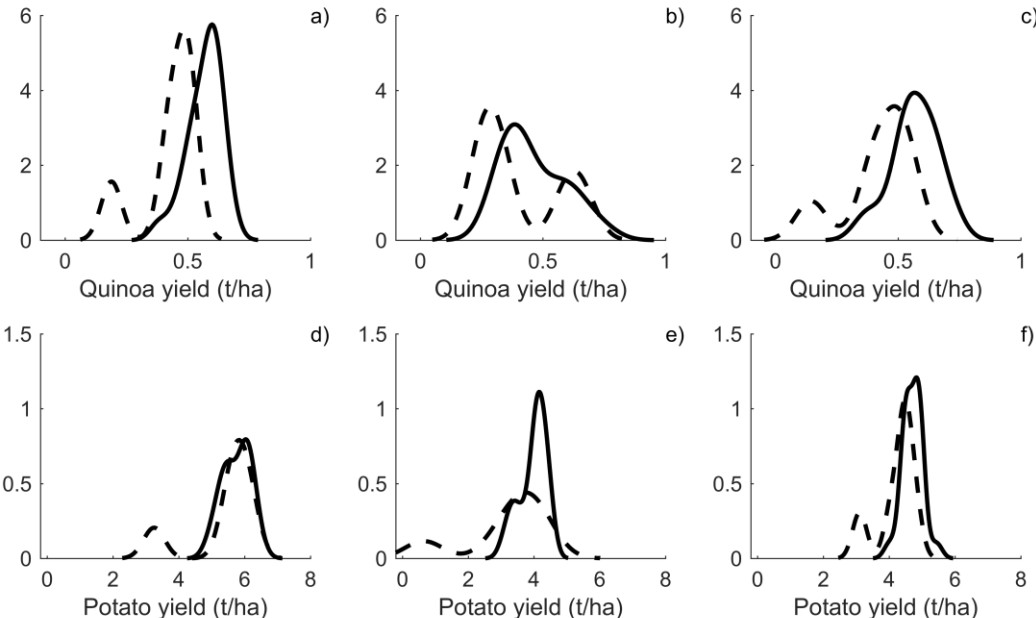

**Figure 3. Density function of quinoa (a, b, c) and potato (d, e, f) yield (t/ha) during El Nino (dashed line) and neutral/moderate (solid line) in (a) and (d) La Paz, (b) and (e) Oruro and (c) and (f) Potosi.**