# Peer review of "Early warning and drought risk assessment for the Bolivian Altiplano agriculture using high resolution satellite imagery data"

_Natural Hazards and Earth System Sciences, 2018_

## Referee Comment (RC1) · Anonymous Referee #1 · 14 Jun 2018

In this article, in-situ observed temperature and precipitation data are compared against satellite data for Bolivia. By distinguishing between El Nino and La Nina years, the impact of this phenomenon on agricultural yield is quantified. In addition, satellite-observed NDVI values are compared against agricutural production data. The authors argue that this analysis contributes to an early warning system for crop deficits.

While the general idea of the study is interesting and the societal importance of an early warning system for crop failure is of obvious importance, the current work leaves much to be desired. The promise of an early warning system in the title is not met, satellite data is used sparingly and the attempt to provide a true risk assessment is

poorly executed. Some of the analyses are interesting, but interpretation of some of these results lacks. Below, these points are explained in more detail.

I am afraid that the current ms. needs considerable work to grow to an acceptable level. My advise to the editor is to reject the ms. in its current form.

Main concerns

1) regarding promisses made in the title: Risk is commonly seen as a combination of 'hazard', 'vulnerability' and 'exposure'. There is a considerable body of literature on this view. The authors have focused on 'hazard' - which is the meteorological aspect of drought (i.e. lack of rain in this case). It is no problem to focus on 'hazard' only - but be clear on this. Also, the 'early warning' aspect of this work is actually nothing more than the statistical relation between ENSO and crop yield, suggesting that when a particular phase of ENSO is forecasted the predictions, then a increase/decrease in precipitation and a response in crop yield is to be expected. Putting forward a statistical relation as 'early warning system' is a bit overdoing it.

2) Satellite-sourced NDVI data are used to compare against crop yield (table 2 and fig. A6). This is an interesting aspect - but it is really something new or unexpected? The other source of satellite data (precipitation from CHIRPS), where do you actually use these data? I see a rather extensive comparison between satellite-based estimates of precipitation and rain gauge data, but where does the satellite data actually enter the analysis? There are no maps of precipitation - which would the minimum to expect when using satellite data.

3) Some results, like those in fig. A6, are interesting. But the question which pops-up immediately when seeing this figure is: why does Oruro behave so much different than the others? This is not discussed and not even observed actually. Similarly, fig. A4 could be interesting as well, but no attempt is made to determine the length and start of the period over which precipitation is accumulated to understand this relation. No motivation is given why the accumulation period is chosen at it is. Also, equation 1 is

introduced, but we see the influence of precipitation only (and not temperature as well).

Other aspects the authors could look into

*) the article is a bit wordy, it could be shortened and made more concise.

*) page 2, line 32. A simple correlation analysis between November-March precipitation and a ENSO index would be great in explaining the relevance of your study. This can be done by using the Climate Explorer (climexp.knmi.nl) using standard available data (or you upload your own data).

*) page 4, line 17. Quantify the amount and length of data gaps you treated by infilling with mean monthly values.

*) I am not very familiar with the CHIRPS data. You write that it is based on the TRMM satellite and the 3B42 product is available since 2000. However, on line 32 of page 4, you claim that the data goes back to 1981. Can you provide a little more explanation?

*) page 5, line 20. Bimonthly means "every 2 months". I guess you mean something like "semimonthly" (at least that is what the Merriam-Webster online dictionary suggests).

*) page 5, lines 27-30. Here you should give a some more detail on the validation of processed NDVI values. Simple questions like: does it reproduce the seasonal cycle? are worth looking into and they give the reader the confidence that this might actually work! For example, a scatter plot of 15-day NDVI values against yearday for a few selected pixels.

*) page 6, line 18. When a correlation exceeding 0.7 is found, I take you label the satellite data as reliable, right? What if this threshold is not reached, what part of the analysis is then not possible?

*) page 6, line 31. Accumulated precipitation and accumulated temperature. Accumulated over what periods? In section 4.2 I read that precipitation is accumulated for 12 months, starting in July. Why is that? Also the start of the accumulation period is

relevant, especially for the degree day metric.

*) page 9, line 4-5. The relative error in the wet season is small because the amount of precipitation is so large - not becuase of small errors in the estimate. Please use a different metric to quantify the accuracy of the satellite-based precipitation.

*) section 4.1 ends with the conclusions that CHIRPS is doing quite well, but the spatial aspect is lacking in this analysis. Are there stations/regions that are over or under estimated? A map with average RMSE valus for January and July would be great here. Note that the distribution of rain gauges is not too homogeneous - so a little caution is in place here.

*) page 10, line 3. Here we see 'mean of maximum annual NDVI' - which is new to me (at least, I did not see it discussed earlier in the article).

*) page 11, line 23-25. Here the text seems to suggest that the 'ENSO warm phase' and 'extreme drought' are two separate things. Obviously, the effects of ENSO is the drought. Please rephrase.

*) fig. A3. Strange that winter values of relative ME and relative RMSE have wide distributions, while the individual months have very narrow distributions. Also: simply leaving-out values (like the monthly June and July values) when they do not fit in the plot, is not done. Include these in the plot and adjust the axis (put in a break - meaning skip some values in the axis - or use a non-linear axis).

---

## Referee Comment (RC2) · Anonymous Referee #2 · 22 Jun 2018

The manuscript concerns an important problem, the development of early warning system for assessment of drought and related yield reduction in agricultural practice. Drought is an important problem for Bolivia region, where the agriculture is one of the main economic activities and drought is an often disaster. Using of satellite information for monitoring risk events and issuing early warnings is relevant and contemporary approach. In general, the work is based on using the standard accumulated growing degree approach in agricultural science and precipitations to assess the yield effects at warm/cold ENSO phases. The authors pretend that high-resolution satellite imagery data are used for precipitation and vegetation characterisation. Quantitative analyses are based on applying statistical correlation methods. Generally, the topic is relevant

for the journal but the quality of the presented material is not satisfactory. There is a discrepancy between the title, the results reported and the conclusion made by the authors. The reported results are not sufficient to support the interpretations and conclusions. The paper is not well conceived and laid down with examples not corroborating the case. The manuscript needs major revision and significant improvement before publishing.

Specific recommendations

1. The title does not reflect the essence of the work presented. Actually there is no "Early warning" of drought because it is only underlined that the early warning of ENSO warm and cold phases is of importance but a methodology of issuing warnings is missing in this work. "Early warning" have to be avoided in the title because it is not approved in the text. # 2. Introduction is too expanded with material, which is beyond the context of the paper. For example: P2/ln 6 'Regarding the prevention . . .These include the setup of insurance and irrigation systems'; the topic is not how to overcome drought but to create a system for drought assessment? P2/ln 25 'Such an approach is now seen as the most 25 appropriate in data-scarce environments (see IPCC, 2012;and UN, 2015)'? Please note that the approach used in the paper here is still not introduced, first motivate the used approach. Instead of this, it is expected the Introduction section to include the state of the art on the specific problem, i.e. on early warning and drought risk assessment as it is declared in the title. Although the scarcity of data in the region, there is some international experience that should be summarized. # 3. P3/ln15 The aim is not relevant presented; it sounds like declaration of something already done? 'We therefore used new precipitation and vegetation satellite data that present full coverage of the spatial distribution in the study area. We combine. . .. . ..'.? Why these data are classified as 'new'? The authors say that this HR satellite quasi-global rainfall dataset is available from 1981 (p4, ln32)? 4. P3 Section 2 'Case Study Region and Data Availability. Again too many words not in the essence of the work Why such details about the population and cities in a scientific paper on

drought are included? Do the readers really need from Table A1 or a map with the location of the stations will be enough and more informative? # 5. P5/ section 2.3 Why crop production and vegetation data in the section title are considered as two different items? Can you provide any finding about the reason for distinguish both. Is it possible to have crop without vegetation? # 6. P5/Ln 17 It is declared that "the Normalized Difference Vegetation Index (NDVI) was used in our study in order to relate climate, vegetation". Actually NDVI is still not used, this is expected to be done in the results section. The author should give proper credit to previous and other related work with clear indication of specific contribution in using NDVI here. # 7. P6/section 3.1. The procedure for validation satellite data using only qualitative 'Yes/No' evaluation is not relevant. For validation purposes of satellite product, rain quantities from the two sources of measurements should be compared, moreover that you are speaking that the 'accuracy' is tested (p.1/Ln 17). Also for the purposes of drought assessment the rain quantity is the essential point. # 8. P6, p.7 Not relevant titles of the section 3.2 and 3.3. please explain why crop yield and vegetation are two different things? # 9. P12 /ln7 The authors should specify which aspect of their work is a first attempt to relate agricultural drought in relation with ENSO? Not enough credit to previous works on this is given. # 10. P12/Ln10 it is claimed '…………, a significant influence of precipitation on vegetation and crop yields in the region was identified'. Everywhere, for each region precipitation has influence on vegetation and crop yield. This cannot be considered as a result from this study. # 11. Ln12 It is claimed by the authors 'Our study provides valuable information for early warning systems, primarily by providing information of the relationship between crop production and vegetation, and subsequently a relation between vegetation and climatological parameters'. What would be the difference between crop production and vegetation if any, please precise terminology so the results to be evident. # 12. There are a lot of references but scare information is used in convincing the results and discussion sections. # 13. The overall presentation is not well structured; there is no relevant balance between results -discussion sections and other parts. The readability of the text is not sufficient.

---

## Short Comment (SC1) · 3 Jul 2018

The manuscript aims at developing an agricultural early warning for crop production in the bolivian Altiplano. Bolivian Altiplano is an arid region where ENSO related drought events may lead to severe economic-social consequences, as it happened in the early 80s and the mid 90s. The manuscript compares some data from meteorological stations with satellite data, and analyzes its correlation with ENSO. Although the data is interesting (and the idea is interesting too), I'm afraid the manuscript has some deficiencies. The manuscript does not propose an early warning system The proposed correlations are too weak to provide confident prediction

Main comments:

1. The term early warning usually refers to a time anticipation; X time before a given event the warning is emitted. In the present case, readers would expect to see the NDVI or ENSO conditions that represent a high probability of drought some months before the drought. For instance, if the September ENSO is X, there is a Y probability of drought. However, the article doesn't show any early warning. The article only analyzes the some ENSO relationships, which is nothing new. Several previous researchers (not referenced in the literature review) have already analyzed the relationship between ENSO and crop production in South America (Izumi et al., 2014; Anderson et al., 2016). Moreover, previous studies have already proposed ENSO based crop management policies in South America (Ramirez-Rodrigues et al., 2014)

2. It is not clear which satellite data was used. The title suggests "high resolution", but the manuscript mentions 0.05 degree (about 5 km). The term high resolution satellite imagery usually refers to less than 90 m or 50 m resolution.

3. The manuscript only shows the results of 1 correlation for 1 station (NDVI correlation for Tihuanacu), which is supposed to be the one with the highest correlation (Pg9 Ln21). This brings several doubts. *How is the correlation in other locations? (Maybe a thematic correlation map would be more appropriate) *Even though Tihuanacu has the highest correlation, the correlation is still quite weak (0.41). Does it means that the other correlations were even lower? *Manuscript states that it considered 23 stations, most them located in La Paz. But then it presents an average value for each department. This procedure is not appropriate for several reasons. For instance: a) It is not fair to compare the average of La Paz (more than 15 stations) with the average of Potosi (only 3 stations poorly distributed) b) In the present methodology the stations were clustered based on political boundaries. Does it mean that bolivian climate behaves according to its political boundaries? I don't believe so. For instance, the stations near Titikaka lake (in La Paz) have very different climatic characteristics than other stations also in La Paz (for instance El Alto or Santiago de Machaca).

*The correlations from FigA6 are different than the values from Table 2 (because of rounding decimals).

*Manuscript states that Tihuanacu has the highest correlation, but Table 2 shows that Potosi has higher correlation.

*The highest correlation (0.51) is still a weak correlation. What is the confidence of a prediction based on such a weak linear correlation? What would be the confidence of predictions for Oruro based on a linear correlation of 0.12? Unfortunately, it seems that the present results suggest that the present methodology does not provide a reliable drought etimatimations.

References: Izumi et al., 2014. Impacts of El Niño Southern Oscillation on the global yields of major crops, Nature Communication

Anderson et al., 2016. Life cycles of agriculturally relevant ENSO teleconnections in North and South America, International journal of climatology, 37, 3297-3318

Ramirez-Rodrigues et al., 2014. Tailoring wheat management to ENSO phases for increased wheat production in Paraguay, Climate risk management, 3, 24-38.

---

## Author Comment (AC1) · 24 Aug 2018

Referee #2 1. i. The title does not reflect the essence of the work presented. Actually there is no "Early warning" of drought because it is only underlined that the early warning of ENSO warm and cold phases is of importance but a methodology of issuing warnings is missing in this work. "Early warning" have to be avoided in the title because it is not approved in the text.

ii. The authors agree and have modified the title of the study to "Drought risk assessment for the Bolivian Altiplano agriculture and its association with El Niño Southern Oscillation". The term "association with ENSO" is more appropriate than "early warning", considering the influence of ENSO on precipitation patterns, in consequence with agricultural production.

2. i. Introduction is too expanded with material, which is beyond the context of the paper. For example: P2/ln 6 'Regarding the prevention . . .These include the setup of insurance and irrigation systems'; the topic is not how to overcome drought but to create a system for drought assessment? P2/ln 25 'Such an approach is now seen as the most 25 appropriate in data-scarce environments (see IPCC, 2012;and UN, 2015)'? Please note that the approach used in the paper here is still not introduced, first motivate the used approach. Instead of this, it is expected the Introduction section to include the state of the art on the specific problem, i.e. on early warning and drought risk assessment as it is declared in the title. Although the scarcity of data in the region, there is some international experience that should be summarized.

ii. The introduction and the second section "Data description" have been shortened. The new text includes only the relevant information for the study, we concisely described the general characteristics of the study area. The section introduction was reduced to include only the important information of the study.

iii. Please see the section Changes in manuscript.

3. i. P3/ln15 The aim is not relevant presented; it sounds like declaration of something already done? 'We therefore used new precipitation and vegetation satellite data that present full coverage of the spatial distribution in the study area. We combine. . .. . ∴? Why these data are classified as 'new'? The authors say that this HR satellite quasi-global rainfall dataset is available from 1981 (p4, ln32)?

ii. The authors agree and have reformulated the text. The aim of the study was reformulated to: "The objective of our paper is to present a methodology for assessing drought risk to mitigate its impacts on agriculture using observed gauge data and satellite based imagery data for the Bolivian Altiplano".

4. i. P3 Section 2 'Case Study Region and Data Availability. Again too many words not in the essence of the work Why such details about the population and cities in a scientific paper on drought are included? Do the readers really need from Table A1 or a map with the location of the stations will be enough and more informative?

ii. The section "Case Study Region and Data Availability", was changed to "Data description". The section was modified and shortened.

iii. Please see the section Changes in manuscript.

5. i. P5/ section 2.3 Why crop production and vegetation data in the section title are considered as two different items? Can you provide any finding about the reason for distinguish both? Is it possible to have crop without vegetation?

ii. There are two main reasons to use NDVI in addition to crop yield for the analysis. One is the spatial resolution of NDVI, that is higher than the available agricultural data during the study period. Crop yield data are available only at local scale (administrative division), NDVI is available at 0.08áţŠ. The second reason to use NDVI is the temporal resolution, NDVI data have a semi-monthly time resolution, allowing to analyze the monthly NDVI average with monthly precipitation total and the monthly temperature average. As discussed currently there are no studies for the Altiplano which relate satellite sourced imagery with agriculture risk, including possible associations with ENSO. The article tries to fill this gap.

iii. This explanation is now included in the manuscript.

6. i. P5/Ln 17 It is declared that "the Normalized Difference Vegetation Index (NDVI) was used in our study in order to relate climate, vegetation". Actually NDVI is still not used, this is expected to be done in the results section. The author should give proper credit to previous and other related work with clear indication of specific contribution in using NDVI here.

ii. The authors agree and the text was modified.

iii. The new text is: "The Normalized Difference Vegetation Index (NDVI) can be used to estimate the vegetation vigour (Ji and Peters, 2003) and crop phenology (Beck et al., 2006). NDVI was assembled from the Advanced Very High Resolution Radiometer (AVHRR) sensors by the Global Inventory Monitoring and Modelling System (GIMMS) at semi-monthly (15 days) time steps with a spatial resolution of 0.08áţŠ. NDVI 3g.v1 (third generation GIMMS NDVI from AVHRR sensors) data set spans from July 1981 to December 2015. Note, the NDVI is an index that presents a range of values from 0 to 1, bare soil values are closer to 0, while dense vegetation has values close to 1 (Holben, 1986). NDVI 3g.v1 GIMMS provides information to differentiate valid values from possible errors due to snow, cloud, and interpolation errors. These errors were eliminated from the dataset and where replaced with the nearest neighbour value. In our study we used the NDVI to simulate crop production".

7. i. P6/section 3.1. The procedure for validation satellite data using only qualitative 'Yes/No' evaluation is not relevant. For validation purposes of satellite product, rain quantities from the two sources of measurements should be compared, moreover that you are speaking that the 'accuracy' is tested (p.1/Ln 17). Also for the purposes of drought assessment the rain quantity is the essential point.

ii. The procedure for validation of satellite data uses both qualitative and quantitative evaluation. To evaluate the satellite product, it was compared with gauged precipitation data. As described in the methods section: "For the quantitative statistics, mean error (ME) or bias, and root mean squared error (RMSE) were calculated based on Wilks (2006), and Nash Sutcliffe efficiency (NSE) coefficient based on Nash and Sutcliffe (1970) was used as well. Spearman rank correlation was used to estimate the goodness of fit to observations, the bias to show the degree of over- or underestimation (Duan et al., 2015), the root mean square error (RMSE) to compute the average magnitude of the estimated errors (values closer to zero generally indicate smaller magnitude of error), and finally the Nash Sutcliffe Efficiency coefficient that evaluates the prediction accuracy compared to observations (1 corresponds to a perfect match

between gauge observation and satellite-based estimate and zero indicates that the satellite estimations are as accurate as the mean of the observed data; negative values indicate that the observed mean is better than satellite-based estimate, see Nash and Sutcliffe (1970) for more details)".

iii. We improved the description of the results obtained in the manuscript. However, we consider that the method used for the satellite validation analysis with a statistical approach is suitable to qualify the accuracy of the estimations, in which the spearman rank correlation, mean error, root mean squared error, and Nash Sutcliffe efficiency were computed.

8. i. P6, p.7 Not relevant titles of the section 3.2 and 3.3. please explain why crop yield and vegetation are two different things?

ii. The relationship between NDVI and crop field was analyzed to show the capability of NDVI for crop yield estimation. Crop yield is represented with the relation of the production (t) and the area (ha) and it is available at major administrative national level in the study area. And the Normalized Difference Vegetation Index (NDVI) can simulate the crop phenology (Beck et al., 2006). In contrast to the crop yield, NDVI can be related with the precipitation and temperature at a monthly time scale and with a spatial resolution of 0.08 degrees.

iii. This information is included in the manuscript.

9. i. P12 /ln7 The authors should specify which aspect of their work is a first attempt to relate agricultural drought in relation with ENSO? Not enough credit to previous works on this is given.

ii. The authors intend to improve the scientific understanding to reduce the risks of drought events on agriculture in association with the warm ENSO phase in regions were climate gauged observations are scare or uneven distributed. The spatial resolution of the satellite precipitation data and the NDVI permit to perform the analysis at the

entire study area. NDVI and crop yield were compared, and the grids with correlations above 0.6 were selected, because they better estimate the crop yield. Afterwards, a linear regression between the selected NDVI grids and the climate variables were developed. The results show the regions were crop yield is more vulnerable to a reduction of precipitation. And a reduction of precipitation is shown due to warm ENSO phases in the study area. With this information, we could define in which areas are more vulnerable to lack of rain, and in consequence to a warm ENSO phase.

Previous studies have found associations between ENSO and precipitation (e.g. Seiler et al., 2013; Garreaud, 2009; Vuille et al., 2000; Francou et al., 2004). Other authors have for example, studied ENSO-based seasonal forecasting (Lupo et al., 2017), and crop relation with climate variability (Garcia et al., 2007; Porter and Semenov, 2005). However, few studies have incorporated remote sensing data to relate drought events and ENSO (e.g. Liu and Juárez, 2001).

iii. This information is now included in the new text.

10. i. P12/Ln10 it is 0 a significant influence of precipitation on vegetation and crop yields in the region was identified'. Everywhere, for each region precipitation has influence on vegetation and crop yield. This cannot be considered as a result from this study.

ii. The authors agree and have reformulated the conclusion "a significant influence of precipitation on vegetation and crop yields in the region was identified". It is obvious that precipitation has an effect on crop yield. The novelty of the study is to improve the drought risk management based on the association of ENSO and crop yield, by using satellite and gauged data.

11. i. Ln12 It is claimed by the authors 'Our study provides valuable information for early warning systems, primarily by providing information of the relationship between crop production and vegetation, and subsequently a relation between vegetation and climatological parameters'. What would be the difference between crop production and

vegetation if any, please precise terminology so the results to be evident.

ii. NDVI can estimate the crop production. The NDVI is an index, and therefore the values are from 0 to 1 [-], in contrast crop yield is the production (t) in relation of the area (ha). In this study the NDVI is used to simulate the crop production, to apply a regression with the climate variables.

iii. The "Conclusions" section was modified (please see the section Changes in manuscript).

12. i. There are a lot of references but scare information is used in convincing the results and discussion sections.

ii. The results and discussion sections were improved (please see the section Changes in manuscript).

13. i. The overall presentation is not well structured; there is no relevant balance between results -discussion sections and other parts. The readability of the text is not sufficient.

ii. Major modifications were included in the manuscript. All the sections of the manuscript were modified. The first section "Introduction" and the second section "Data Description" where shortened to the relevant information. The third section "Methods" was modified as well, a new analysis of the relationship of the NDVI and the crop yield was done. The section with major changes is "Results", because we incorporated the new results of the relationship between NDVI and crop yield, and the new regression of NDVI and climate variables The discussion was also modified to suit with the new aim and findings. Finally the las section "Conclusions" were adapted to the new manuscript content.

iii. The new text is described in Changes in Manuscript.

Changes in manuscript

In general, there was an improvement in the manuscript redaction and structure. The title was modified to "Drought risk assessment for the Bolivian Altiplano agriculture and its association with El Niño Southern Oscillation". The new objective of our paper is to present a methodology for assessing drought risk to mitigate its impacts on agriculture using observed gauge data and satellite based imagery data for the Bolivian Altiplano. The description of the modifications of the manuscript will be dived by sections: 1) Introduction, 2) Data Description, 3) Methods, 4) Results, 5) Discussion and, 6) Conclusion. The sections 1) Introduction and 2) Data Description where shortened to the relevant information. The section 3) Methods was modified as well, a new analysis of the relationship of the NDVI and the crop yield was done. The section with major changes is 4) Results, because we incorporated the new results of the relationship between NDVI and crop yield, and the new regression of NDVI and climate. Finally, the section 5) Discussion was also modified to suit with the new aim and findings, and 6) Conclusions" were adapted to the new manuscript content.

1. Introduction

Agricultural production is highly sensitive to weather extremes, including droughts and heat waves. Losses due to such extreme hazard events pose a significant challenge to farmers as well as governments worldwide (UNISDR, 2015). Worryingly, the scientific community predicts an amplification of these negative impacts due to future climate change (IPCC, 2013). Especially in developing countries as Bolivia, drought is as a major natural hazard. However, the impacts vary on a seasonal and annual timescale, on the hazard intensity, and the capacity to prevent and respond to droughts (UNISDR, 2009). Regarding the former, the El Niño Southern Oscillation (ENSO) plays an important role. ENSO triggers droughts in several regions around the world, driving losses in agricultural crops and increased food insecurity (Kogan and Guo, 2017).

Bolivia have experienced large socio-economic losses in the past due to droughts. Generally, agricultural productivity in the Bolivian Altiplano is low due to high susceptibility to the climate, poor soil conditions, and the mainly manual labour. Poor agricultural production is also associated with the ENSO climate phenomena (Buxton et al., 2013). Droughts are generally driven by the ENSO warm phases (Vicente-Serrano et al., 2015; Garreaud and Aceituno, 2001; Thompson et al., 1984). Most important rainfed crops in the region include quinoa and potato. The Sustainable Development Goals (SDGs) state that the priority areas for adaptation to climate change are water and agriculture. These in turn, are related to the largest climate hazards including floods, droughts, and higher temperatures (UN, 2016). Additionally, the implementation of early warning is fundamental for drought disaster risk management, proactive planning, and mitigation policy measures in vulnerable regions, including Latin American countries such as Bolivia (Verbist et al., 2016).

Various studies of the relationship between ENSO and precipitation were developed previously, and they show a negative relationship between ENSO warm phase (El Nino) and precipitation, meaning that El Nino periods have been linked with a decrease of precipitation (Vicente-Serrano et al., 2015; Garreaud and Aceituno, 2001; Thompson et al., 1984; Francou et al., 2004; Vuille et al., 2000; Vuille, 1999). Other authors have studied ENSO-based seasonal forecasting (Lupo et al., 2017), and crop relation with climate variability (Garcia et al., 2007; Porter and Semenov, 2005). On the other hand, previous research of the relationship between ENSO and crop production was developed at global scale (Iizumi et al., 2014) and in South America (Anderson et al., 2017). Moreover, previous studies have already proposed ENSO based crop management in South America (Ramirez-Rodrigues et al., 2014). However, few studies have incorporated remote sensing data to relate drought events and ENSO (e.g. Liu and Juárez, 2001), and they are not developed in the Altiplano. In consequence, there is a gap of a drought risk assessment for the Bolivian Altiplano agriculture and its association with El Niño Southern Oscillation, using observed gauge data and satellite based imagery data. To lessen the long term impacts of these events, the national government has allocated a large budget for emergency operations to compensate part of the losses, which are usually evaluated in an ex-post approach. However, based on ENSO forecasting, an El Nino event can be predicted 1 to 7 months ahead (Tippett et al., 2012).

For this time period it may be possible to implement ex-ante policies to reduce societal vulnerability to droughts, stressing preparedness, and improve risk management strategies. We are especially interested in how a risk based approach can be used to determine the potential need of resources as well as ways to determine hotspots where it is likely that these resources need to be distributed. A constraint to study drought occurrence is the uneven and scarce distribution of weather and crop related ground data in the region. We therefore use precipitation and vegetation satellite data that present full coverage of the spatial distribution in the study area. We combine this information with gauged precipitation, temperature, and crop yield data to enhance the knowledge and provide consistent results for climate and vegetation variability. Based on these observations, the objective of our paper is to present a methodology for assessing drought risk to mitigate its impacts on agriculture using observed gauged data and satellite based imagery data for the Bolivian Altiplano.

Our paper is organized as follows. Given the importance of data limitations in our case study region, we first introduce in section 2 the recently available datasets employed for our analysis. In section 3 we discuss the methods employed to (i) test the validity of the new datasets in our analysis, (ii) show how the relationship between climate and vegetation data was estimated, and (iii) how ENSO was incorporated in our analysis. After this, section 4 presents results and our proposed framework. Finally, section 5 ends with a conclusion, ways forward, and outlook to the future.

2. Data Description

The Bolivian Altiplano covers about 150,000 km2. It contains more than 70% of the total Altiplano surface area, the remaining percentage is located in southern Peru and northern Chile. La Paz, Oruro, and Potosi are the major administrative regions in the Bolivian Altiplano. The Altiplano has a pronounced southwest-northeast gradient (200–900 mm year$-1$) in annual precipitation with the wet season occurring from November to March (Garreaud et al., 2003). Over 60% of the annual precipitation occur during the summer months (DJF) in association with the South American Monsoon (SAM) (see

Fig. A1).

**2.1 Climate Variables: Precipitation and Temperature Data**

Time series of observed monthly precipitation at 23 locations and mean, maximum, and minimum temperature at 15 locations from July 1981 to June 2016 were obtained from the National Service of Meteorology and Hydrology (SENAMHI; see Appendix Table A1). Initially, the available precipitation data sets included 65 gauges but only 23 were used since they have less than 10% of missing data. The data gaps were filled with the mean monthly value of the whole dataset to provide full time series. Outliers were identified by comparing with neighbouring monthly data. The inter-annual temperature at the 15 locations varied considerably between summer (DJFM) and winter (JJAS), including a larger variance for the minimum temperature (Fig. A2a). Regions close to the Lake Titicaca present lower inter-annual variability (Copacabana [10], Fig. A2b). In contrast, Uyuni [22] presents larger inter-annual oscillations (Fig. A2c). The precipitation gauges have an uneven spatial distribution and are mainly concentrated in the northern Bolivian Altiplano. To improve the spatial coverage of rainfall data, monthly quasi-rainfall time series from satellite data were included in our study. The Climate Hazards Group InfraRed Precipitation with station data (CHIRPS) quasi-global rainfall dataset was used. CHIRPS presents a 0.05° resolution satellite imagery and is a quasi-global rainfall dataset from 1981 to the near present with a satellite resolution of 0.05° (Funk et al., 2015). The information about CHIRPS is described in http://chg.geog.ucsb.edu/data/chirps/.

**2.2 El Niño Southern Oscillation Data**

The Oceanic Niño Index (ONI) is usually used to identify El Niño (warm) and La Niña (cool) years (http://www.cpc.ncep.noaa.gov/). ONI is the 3 month running mean of Extended Reconstructed Sea Surface Temperature (ERSST v5) anomalies in the El Niño 3.4 region. The El Niño 3.4 anomalies represent the average equatorial SSTs in the equatorial Pacific Ocean (5oN to 5oS latitude, and 120o to 170oW longitude). Five

consecutive overlapping three month periods at or above +0.5°C anomaly represents warm events (El Niño), and at or below the -0.5 anomaly are cold (La Niña) events.

2.3 Crop Production and Vegetation Data

As indicated above, quinoa and potato are the main crops in the Bolivian Altiplano and still gaining importance. The quinoa growing season is from September to April and, for potato from October to March. Data for quinoa and potato yield were obtained from the National Institute of Statistics (INE) of Bolivia from July 1981 to June 2016 for La Paz, Oruro, and Potosi (Fig. 1). No crop yield data at local scale are available and this is a major limitation that needs to be addressed in the future. The annual datasets represent production (t) in relation of the area (ha) at regional level.

Additionally, the Normalized Difference Vegetation Index (NDVI) can be used to estimate the vegetation vigour (Ji and Peters, 2003) and crop phenology (Beck et al., 2006). NDVI was assembled from the Advanced Very High Resolution Radiometer (AVHRR) sensors by the Global Inventory Monitoring and Modelling System (GIMMS) at semi-monthly (15 days) time steps with a spatial resolution of 0.08áţŠ. NDVI 3g.v1 (third generation GIMMS NDVI from AVHRR sensors) data set spans from July 1981 to December 2015. Note, the NDVI is an index that presents a range of values from 0 to 1, bare soil values are closer to 0, while dense vegetation has values close to 1 (Holben, 1986). NDVI 3g.v1 GIMMS provides information to differentiate valid values from possible errors due to snow, cloud, and interpolation errors. These errors were eliminated from the dataset and where replaced with the nearest neighbour value. In our study we used the NDVI to simulate crop production.

3. Methods

3.1 NDVI simulation of crop yield

The maximum 15-days NDVI of March, April and May for every year was identified. Only the time from March to May was considered because this period represents the
maximum phenological development of quinoa and potato crops. The maximum NDVI of each grid was compared to the annual crop yield at La Paz, Oruro, and Potosi. The NDVI grids and crop yield correlations larger than 0.6 (spearman correlation, p = 0.001) were considered as adequate for crop yield estimation, and only these grids were considered for further study. A similar approach was used by (Huang et al., 2014). Afterwards, a regression of the selected NDVI grids and the precipitation was developed. The satellite precipitation data was used for the regression. For this analysis, the NDVI grids were compared to the same spatial location of satellite precipitation data.

4. Results

4.1 Validation of Chirps satellite precipitation data

The mean annual gauged precipitation and CHIRPS satellite data product (Fig. R1) shows the relevance of the application of satellite data in the studied region.

The RMSE and ME (Fig. R5) shows the locations where satellite data overestimates or underestimates the gauged precipitation. Generally, the precipitation is under/overestimated in a range of -10 to +10 mm per month [6] (Fig. R5). However Charazani [6] present a large bias of 40.7 mm/month. As well, most of the stations present a RMSE between 15 to 30 mm/month, Charazani [6] had a RMSE of 58 mm/month

4.2 NDVI simulation of crop yield

Figure R6 presents the locations where NDVI simulates the crop yield with a correlation larger than 0.6 (spearman correlation, p = 0.001). The NDVI grids that better simulates the quinoa crop are shown in Fig. R6a, and the NDVI that better simulates the potato crop are shown in Fig. R6b. We can see that NDVI can simulate the proper production of the crop land area in La Paz, Oruro, and Potosi. 4.3 NDVI association with precipitation

Only the NDVI grids with larger correlation of 0.6 with crop yield were considered for

the climate regression. The results for stepwise linear regression between NDVI and accumulated precipitation were statistically significant at the 0.01 level. The analysis was firstly applied with the NDVI that best simulates the quinoa yield and afterwards with the NDVI that best simulates the potato yield. In La Paz, the mean correlation coefficient resulting from the regression of the NDVI that best simulates the quinoa and accumulated precipitation is 0.7 in La Paz, and above 0.6 in Oruro and Potosi. The regression of the NDVI that best estimates the potato yield and the accumulated precipitation shows a mean correlation coefficient of 0.7 in La Paz, 0.6 in Oruro, and 0.5 in Potosi.

4.4 Analysis of the ENSO impact on crop yield

Large reductions of crop yield are shown during El Nino events. The crop yield difference between the mean of the crop yield from 1981 to 2016 and the crop yield during a strong El Nino event 1982/1883 is shown in Fig R7. Quinoa yield reduces 65%, 10% and 73% at La Paz, Oruro, and Potosi respectively. And potato yield reduces 42%, 83% and 32% at La Paz, Oruro, and Potosi. In addition, the comparison between the mean NDVI from 1981 to 2016 and the NDVI during El Nino event of 1982-1983 is shown in Fig R8. Around the Lake Titicaca in the northern Altiplano the NDVI reduction is from 0 to 10%. And, in some locations of the southern Altiplano the reductions of NDVI are above 30%, and few locations reaches reductions above 40%, meaning great losses.

Fig. R8 NDVI reduction between the NDVI during a strong El Nino 1982-1983 and the mean NDVI from 1981 to 2016 for a) quinoa and b) potato crop.

4.5 Discussion

We employed a satellite product dataset and tested it for accuracy as well as performance to similar (but with coarser resolution) datasets available for our region. Using this dataset, it was shown that during El Nino years the crop yield reduces (Fig. R7), and as a consequence the socio-economic vulnerability of the farmers increases. Furthermore, it was found that NDVI can be related to crop yield and therefore, NDVI could

be used to target specific hot spots depending on NDVIs availability at a local scale. As a consequence, ENSO forecasts as well as possible magnitudes of crop deficits could be established by the authorities, including identification of possible hotspots of crop deficits during the growing season. Our approach therefore, can not only help for determining the magnitude of assistance needed for farmers at the local level but also enable a pro-active approach to disaster risk management against droughts. This may include not only economic related instruments such as insurance but also risk reduction instruments such as irrigation needs. In fact, early warning based financing is gaining increasing attraction in some real world settings as it has several advantages. However, it should be acknowledged that large challenges still remains (French and Mechler, 2017). Drought severity could be measured via shifts from normal conditions of climatic parameters such as precipitation. As in our case, we not only provided shifts but the difference in risk for El Nino and neutral/moderate years. However, one of the main challenges of drought risk analysis is data-scarcity, e.g., low density or not evenly distributed stations for hydro-meteorological data networks, poor data quality due to missing data, and restricted use of data between government agencies or other institutions. As it was shown here, ENSO warm phase related characteristics are especially important in the context of extreme drought events and should therefore be incorporated within early warning systems as standard practice. Despite these challenges for development of drought risk assessment, applications have been successful in the past. There are numerous cases in many countries around the globe. As in our case, particularly in the mid-latitudes weather patterns are strongly influenced by ENSO. Monitoring and predicting ENSO can therefore significantly contribute to reduce the risk of disasters.

4.6 Conclusions

This study is a first attempt to provide an agricultural drought risk assessment in relation to the ENSO phenomenon for the Bolivian Altiplano. Given the large differences in risk, and corresponding strategies to lessen the impacts could be implemented in the

Bolivian Altiplano. In doing so, we introduced and tested a satellite product that was used for estimating crop risk. The ENSO impact on crop production was evaluated by studying the relation of crop yield, vegetation and climate variables, considering that El Nino generally drives a drought event. Our study provides valuable information for drought risk reduction, primarily by providing information of the hotspots where crop yield is more affected for droughts. Moreover, we showed that ENSO phases are strongly related to crop yield, and with this information the prediction of ENSO could be used to define risks in terms of decrease in crop yields in the studied region. While overall good fit among climate, ENSO, and crop yield variables were found, it is important to consider other parameters, such as evapotranspiration and soil moisture in improved models. With such information also agricultural models could be set up and risk management plans with better accuracy determined.

References

Anderson, W., Seager, R., Baethgen, W., and Cane, M.: Life cycles of agriculturally relevant ENSO teleconnections in North and South America, Int. J. Climatol., 37, 3297-3318, doi:10.1002/joc.4916, 2017.

Beck, P. S. A., Atzberger, C., Høgda, K. A., Johansen, B., and Skidmore, A. K.: Improved monitoring of vegetation dynamics at very high latitudes: A new method using MODIS NDVI, Remote Sens. Environ., 100, 321-334, https://doi.org/10.1016/j.rse.2005.10.021, 2006.

Buxton, N., Escobar, M., Purkey, D., and Lima, N.: Water scarcity, climate change and Bolivia: Planning for climate uncertainties, Stockholm Environment Institute U.S. Center – Davis Office Davis, USA, 4, 2013.

Duan, Y., Wilson, A. M., and Barros, A. P.: Scoping a field experiment: error diagnostics of TRMM precipitation radar estimates in complex terrain as a basis for IPHEx2014, Hydrol. Earth Syst. Sci., 19, 1501-1520, 10.5194/hess-19-1501-2015, 2015.

[Figure]

Francou, B., Vuille, M., Favier, V., and Cáceres, B.: New evidence for an ENSO impact on low-latitude glaciers: Antizana 15, Andes of Ecuador, 0°28′S, J. Geophys. Res. Atmos., 109, doi:10.1029/2003JD004484, 2004.

French, A., and Mechler, R.: Managing El Niño Risks Under Uncertainty in Peru: Learning from the past for a more disaster-resilient future, International Institute for Applied Systems Analysis, Laxenburg, Austria, 2017.

Funk, C., Peterson, P., Landsfeld, M., Pedreros, D., Verdin, J., Shukla, S., Husak, G., Rowland, J., Harrison, L., Hoell, A., and Michaelsen, J.: The climate hazards infrared precipitation with stations—a new environmental record for monitoring extremes, A Nature Research Journal, 2, 150066, 10.1038/sdata.2015.66, 2015.

Garcia, M., Raes, D., Jacobsen, S. E., and Michel, T.: Agroclimatic constraints for rainfed agriculture in the Bolivian Altiplano, J Arid Environ., 71, 109-121, https://doi.org/10.1016/j.jaridenv.2007.02.005, 2007.

Garreaud, R., Vuille, M., and Clement, A. C.: The climate of the Altiplano: observed current conditions and mechanisms of past changes, Palaeogeogr. Palaeoclimatol. Palaeoecol., 194, 5-22, 10.1016/S0031-0182(03)00269-4, 2003.

Garreaud, R. D., and Aceituno, P.: Interannual rainfall variability over the South American Altiplano, J. Clim., 14, 2779-2789, 10.1175/1520-0442(2001)014<2779:Irvots>2.0.Co;2, 2001. Garreaud, R. D.: The Andes climate and weather, Adv. Geosci., 22, 3-11, 10.5194/adgeo-22-3-2009, 2009.

Holben, B. N.: Characteristics of maximum-value composite images from temporal AVHRR data, Int J Remote Sens, 7, 1417-1434, 10.1080/01431168608948945, 1986.

Huang, J., Wang, H., Dai, Q., and Han, D.: Analysis of NDVI Data for Crop Identification and Yield Estimation, IEEE Journal of Selected Topics in Applied Earth Observations and Remote Sensing, 7, 4374-4384, 10.1109/JSTARS.2014.2334332, 2014.

Iizumi, T., Luo, J.-J., Challinor, A. J., Sakurai, G., Yokozawa, M., Sakuma, H.,

Brown, M. E., and Yamagata, T.: Impacts of El Niño Southern Oscillation on the global yields of major crops, Nature Communications, 5, 3712, 10.1038/ncomms4712 https://www.nature.com/articles/ncomms4712#supplementary-information, 2014.

IPCC: Climate Change 2013: The Physical Science Basis. Contribution of Working Group I to the Fifth Assessment Report of the Intergovernmental Panel on Climate Change, edited by: Stocker, T. F., Qin, D., Plattner, G.-K., Tignor, M., Allen, S. K., Boschung, J., Nauels, A., Xia, Y., Bex, V., and Midgley, G. F., Cambridge University Press, Cambridge, United Kingdom and New York, NY, USA, 1535 pp., 2013.

Ji, L., and Peters, A. J.: Assessing vegetation response to drought in the northern Great Plains using vegetation and drought indices, Remote Sens. Environ., 87, 85-98, https://doi.org/10.1016/S0034-4257(03)00174-3, 2003.

Kogan, F., and Guo, W.: Strong 2015–2016 El Niño and implication to global ecosystems from space data, Int J Remote Sens, 38, 161-178, 10.1080/01431161.2016.1259679, 2017.

Liu, W. T., and Juárez, R. I. N.: ENSO drought onset prediction in northeast Brazil using NDVI, Int J Remote Sens, 22, 3483-3501, 10.1080/01431160010006430, 2001.

Lupo, A. R., Garcia, M., Rojas, K., and Gilles, J.: ENSO Related Seasonal Range Prediction over South America, Proceedings, 1, 682, 2017.

Nash, J. E., and Sutcliffe, J. V.: River flow forecasting through conceptual models part I — A discussion of principles, J. Hydrol., 10, 282-290, https://doi.org/10.1016/0022-1694(70)90255-6, 1970.

Porter, J. R., and Semenov, M. A.: Crop responses to climatic variation, Philosophical Transactions of the Royal Society B: Biological Sciences, 360, 2021-2035, 10.1098/rstb.2005.1752, 2005.

Ramirez-Rodrigues, M. A., Asseng, S., Fraisse, C., Stefanova, L., and Eisenkolbi, A.: Tailoring wheat management to ENSO phases for increased wheat production in Paraguay, Climate Risk Management, 3, 24-38, https://doi.org/10.1016/j.crm.2014.06.001, 2014.

Seiler, C., Hutjes, R. W. A., and Kabat, P.: Climate Variability and Trends in Bolivia, J. Appl. Meteorol. Clim., 52, 130-146, 10.1175/Jamc-D-12-0105.1, 2013.

Thompson, L. G., Mosley-Thompson, E., and Arnao, B. M.: El Nino-Southern Oscillation events recorded in the stratigraphy of the tropical Quelccaya ice cap, Peru, Science, 226, 50-53, 10.1126/science.226.4670.50, 1984.

Tippett, M. K., Barnston, A. G., and Li, S.: Performance of Recent Multimodel ENSO Forecasts, J. Appl. Meteorol. Clim., 51, 637-654, 10.1175/jamc-d-11-093.1, 2012.

UN: The Sustainable Development Goals Report 2016, United Nations New York, USA, 2016. UNISDR: Drought Risk Reduction Framework and Practices: Contributing to the Implementation of the Hyogo Framework for Action, United Nations secretariat of the International Strategy for Disaster Reduction (UNISDR), Geneva, Switzerland, 2009.

UNISDR: Making Development Sustainable: The Future of Disaster Risk Management. Global Assessment Report on Disaster Risk Reduction, United Nations Office for Disaster Risk Reduction (UNISDR), Geneva, Switzerland, 266, 2015.

Verbist, K., Amani, A., Mishra, A., and Cisneros, B. J.: Strengthening drought risk management and policy: UNESCO International Hydrological Programme's case studies from Africa and Latin America and the Caribbean, Water Policy, 18, 245-261, 10.2166/wp.2016.223, 2016.

Vicente-Serrano, S. M., Chura, O., López-Moreno, J. I., Azorin-Molina, C., Sanchez-Lorenzo, A., Aguilar, E., Moran-Tejeda, E., Trujillo, F., Martínez, R., and Nieto, J. J.: Spatio-temporal variability of droughts in Bolivia: 1955–2012, Int. J. Climatol., 35, 3024-3040, 10.1002/joc.4190, 2015.

Vuille, M.: Atmospheric circulation over the Bolivian Altiplano during dry and wet periods and extreme phases of the Southern Oscillation, Int. J. Climatol., 19, 1579-1600,

10.1002/(SICI)1097-0088(19991130)19:14<1579::AID-JOC441>3.0.CO;2-N, 1999.

Vuille, M., Bradley, R. S., and Keimig, F.: Interannual climate variability in the Central Andes and its relation to tropical Pacific and Atlantic forcing, J. Geophys. Res. Atmos., 105, 12447-12460, 10.1029/2000JD900134, 2000.

Wilks, D. S.: Statistical Methods in the Atmospheric Sciences, second ed., Academic Press, 2006.

Please also note the supplement to this comment:
https://www.nat-hazards-earth-syst-sci-discuss.net/nhess-2018-133/nhess-2018-133-AC1-supplement.pdf

———————————————————————

[Figure]

**Supplement:**

[Figure]

Fig. R1. Map of mean annual precipitation (July 1981- June 2016) of (a) gauged data* and isohyets** and (b) CHIRPS satellite product. Source: *SENAMHI, **Ministry of Rural Development and Land of Bolivia.

[Figure]

Fig R2. Scatterplot of the day of the year (DOY) and 15-day NDVI from July 1981 to December 2015 for the 23 locations described in Table A1.

[Figure]

Figure R3. Scatterplot of monthly gauged and satellite precipitation data for the 23 studied locations from July 1981 to December 2015.

[Figure]

Fig. R4. RMSE for (a) January and (b) June.

[Figure]

Fig. R5. Map of the Altiplano showing (a) RMSE and (b) ME at the 23 studied locations from July 1981 to June 2016

[Figure]

Fig R6. Correlation coefficient (R2) of the regression of NDVI as the predictand, and precipitation as the predictor for the grids where NDVI better estimate the (a) quinoa and (b) potato yield.

[Figure]

Fig. R7. Comparison of the crop production between the mean crop yield from 1981 to 2016 and the crop yield of 1982/1983 of a) quinoa and b) potato crops.

[Figure]

Fig. R8 NDVI reduction between the NDVI during a strong El Nino 1982-1983 and the mean NDVI from 1981 to 2016 for a) quinoa and b) potato crop.

---

## Author Comment (AC2) · 24 Aug 2018

Response to reviewers

The authors thank the editor, reviewers, and third person's comments for time spent and efforts to improve the manuscript. We have revised the manuscript accordingly taking the reviewers' and third person's comments into due consideration. Below follow answers to the reviewer's comments and description of actions taken.

While the reviewers indicated that the article is of substantial interest and relevant for the journal they criticized the misleading title as well as the analysis to be "poorly exe-

cuted". Furthermore, they reviewers indicated that the quality of the presented material is "not satisfactory" to support the results found. We agree with the reviewers in the sense that one major restriction to provide a sophisticated model between satellite imagery and agriculture risk is data limitation. However, the unavailability of data to be used for a sophisticated drought modelling approach are very common in a developing country context (World Bank 2016). One way to overcome this challenge is to apply a so-called iterative risk management approach, e.g. starting with baseline estimates using the best data currently available and updating risk estimates continuously over time (see IPCC 2012). Currently there are no studies for the Altiplano which relates satellite sourced imagery with agriculture risk, including possible associations with ENSO. The article tries to fill this gap. We acknowledge the fact that there are other studies in other countries which are using more sophisticated models, however, the situation in the Altiplano, especially the data limitations, restricts the use of such models and we provide a way forward to improve the data situation using high-resolution satellite imagery with a probabilistic approach for agriculture risk. Hence, for the current situation in Bolivia our approach can be regarded as one way forward, and can be used as a baseline case for further analysis in the future. As indicated, the situation is quite similar in other developing countries around the world and our approach can be seen as one way forward how to implement drought risk management under data scarcity including the important connection with the ENSO phenomenon. We also provided now much more detail on the strengths and limitations of the approach, including a detailed uncertainty analysis. Please find our detailed responses to the reviewer comments below.

The response to reviewers are structured following the recommendations of the editors:
i. Comment from referee

ii. Authors' response

iii. Author's changes in manuscript.

In addition the last section "Changes in manuscript" describes in more detail the author's changes in manuscript.

Main concerns

1) i. Regarding promises made in the title: Risk is commonly seen as a combination of 'hazard', 'vulnerability' and 'exposure'. There is a considerable body of literature on this view. The authors have focused on 'hazard' - which is the meteorological aspect of drought (i.e. lack of rain in this case). It is no problem to focus on 'hazard' only - but be clear on this. Also, the 'early warning' aspect of this work is actually nothing more than the statistical relation between ENSO and crop yield, suggesting that when a particular phase of ENSO is forecasted the predictions, then an increase/decrease in precipitation and a response in crop yield is to be expected. Putting forward a statistical relation as 'early warning system' is a bit overdoing it.

ii. The authors agree and have modified the title of the study to "Drought risk assessment for the Bolivian Altiplano agriculture and its association with El Niño Southern Oscillation". The manuscript focus now more on insights for the case study determining the drought impact on agriculture by studying the crop production reduction due to drought events. These reductions are related to losses that farmers experience due to this hazard, and the text now identifies hotspots where drought event effects are larger compared to other regions. As we calculate the damages of drought, we propose that the term risk is appropriate. On the other hand, considering the analysis of the influence of ENSO on agricultural production, the term "association with ENSO" is reflecting better the analysis compared to the term "early warning".

iii. The new title of the manuscript is: "Drought risk assessment for the Bolivian Altiplano agriculture and its association with El Niño Southern Oscillation".

The new aim of the manuscript is: "The objective of our paper is to present a methodology for assessing drought risk to mitigate its impacts on agriculture using observed gauge data and satellite based imagery data for the Bolivian Altiplano".

The firsts and second sections of the article were shortened (please have a look to the section Changes in manuscript).

2) i. Satellite-sourced NDVI data are used to compare against crop yield (table 2 and fig. A6). This is an interesting aspect - but it is really something new or unexpected? The other source of satellite data (precipitation from CHIRPS), where do you actually use these data? I see a rather extensive comparison between satellite-based estimates of precipitation and rain gauge data, but where does the satellite data actually enter the analysis? There are no maps of precipitation - which would the minimum to expect when using satellite data.

ii. Currently there are no studies for the Altiplano which relate satellite sourced imagery with agriculture risk, including possible associations with ENSO. The article attempts to fill this gap. We acknowledge the fact that there are other studies in other countries which are using more sophisticated models, however, the situation in the Altiplano, especially the data limitations, restricts the use of such models and we provide a way forward to improve the data situation using satellite imagery with a probabilistic approach for agriculture risk. For the current situation in Bolivia, our approach can be used as a baseline for further analysis in the future (this is also in line with the iterative risk management approach for developing countries explained in the IPCC SREX report, see also Mochizuki et al. 2016). Regarding our approach, in more detail NDVI is a vegetation index and for this reason it relates positively with crop production. There are two main reasons to use NDVI in addition to crop yield for the analysis. One is the higher spatial resolution of NDVI. This is because the agricultural data are only available for major administrative regions during the study period. The NDVI cell grid is 0.08áțŠ. The second reason to use NDVI is that the temporal resolution of 15 days (semi-monthly), allows to analyze the monthly NDVI average and compare it with the monthly total precipitation and mean temperature.

Precipitation data gauged and CHIRPS datasets (Fig. R1) were used to determine the linear regression between vegetation and climate variables. As explained in the

manuscript, there are large limitations for the observed data availability. For instance, gauged data at only 23 locations could be used in the study due to the limited quality of the time series. However, the datasets selected had less than 10% data gaps. The relation between gauged and satellite data are generally higher than 0.7 (spearman correlation at a significance of 0.001). This finding could provide insights to improve the reliability of satellite data, and reduce the limitations of the gauged data availability. Thus, this is an important finding of the paper.

iii. A map of precipitation was elaborated. Figure R1a shows that the gauged stations are concentrated to the northeast where the average annual precipitation is above 500 mm. In contrast the south has a low concentration of stations, where the mean annual precipitation is lower than 300 mm. Figure R2b illustrates the precipitation gradient from northeast to southwest using CHIRPS satellite data product.

3) i. Some results, like those in fig. A6, are interesting. But the question which pops-up immediately when seeing this figure is: why does Oruro behave so much different than the others? This is not discussed and not even observed actually. Similarly, fig. A4 could be interesting as well, but no attempt is made to determine the length and start of the period over which precipitation is accumulated to understand this relation. No motivation is given why the accumulation period is chosen at it is. Also, equation 1 is introduced, but we see the influence of precipitation only (and not temperature as well).

ii. Figure A6 shows the relation of NDVI and crop yield at regional level (La Paz, Oruro, and Potosi). A modified analysis of the relationship between these two variables was developed. Before, the mean NDVI maximum was calculated by using the average of the maximum annual NDVI at La Paz, Oruro and Potosi. To avoid potential errors, the maximum 15-day NDVI of March, April and May for every year was identified. Only the period from March to May was considered because this represents the maximum phenological development of quinoa and potato crops. The maximum NDVI of each grid was compared to the annual crop yield at La Paz, Oruro and Potosi. Consequently, the average of maximum NDVI was not used in the new manuscript. The results of this

analysis are explained in the new text.

Figure A4 shows the accumulated precipitation in relation to NDVI. The accumulated precipitation period was from July to June. This period represents the precipitation cycle of the study region. The rainy season lasts from December to February. This period concentrates about 70% of the total annual precipitation (Garreaud et al., 2003). Figure A2 and A5 show that the lowest temperatures occur during June and July, and the highest temperature in December. The accumulated degree-day period starts in July and ends in June. It was analyzed with the accumulated precipitation period to include the entire growing period of quinoa (September-April) and potato (October-March) crops.

iii. The method and results of the relationship between NDVI and crop production are explained in the section Changes in manuscript.

The motivation for the selected accumulated period for precipitation and temperature are now explained in the manuscript.

Other aspects the authors could look into

*) i. the article is a bit wordy, it could be shortened and made more concise.

ii. The introduction and the second section "Data description" have been shortened. The new text includes only the relevant information for the study, we concisely described the general characteristics of the study area.

iii. The new text is written in the section: Changes in manuscript.

*) i. page 2, line 32. A simple correlation analysis between November-March precipitation and a ENSO index would be great in explaining the relevance of your study. This can be done by using the Climate Explorer (climexp.knmi.nl) using standard available data (or you upload your own data).

ii. The reviewers suggested an analysis between ENSO and precipitation. The authors
agree with the relevance of this analysis, however, a correlation analysis is not able to show a complete picture of the complex relationship. Various studies of the relationship between ENSO and precipitation were developed previously, and they show a negative relationship between ENSO warm phase (El Nino) and precipitation, meaning that El Nino periods have been linked with a decrease of precipitation (Vicente-Serrano et al., 2015; Garreaud and Aceituno, 2001; Thompson et al., 1984; Francou et al., 2004; Vuille et al., 2000; Vuille, 1999). In addition, the authors studied the relationship between precipitation and climate indices including ENSO. The findings of this study are in agreement with the previously mentioned studies. These results have been submitted to the International Journal of Climatology.

iii. This information is now included in the manuscript.

*) i. page 4, line 17. Quantify the amount and length of data gaps you treated by infilling with mean monthly values.

ii. From the available 65 precipitation gauges, 23 presented less than 10% of data gaps. These 23 data series were used in the study. The data gaps were filled with the mean monthly value.

iii. We included this information in the text.

*) i. I am not very familiar with the CHIRPS data. You write that it is based on the TRMM satellite and the 3B42 product is available since 2000. However, on line 32 of page 4, you claim that the data goes back to 1981. Can you provide a little more explanation?

ii. The CHIRPS involves three components: 1) the Climate Hazards group Precipitation climatology (CHPclim), 2) the satellite-only Climate Hazards group Infrared Precipitation (CHIRP), and 3) a station data blending. Firstly, the CHPclim includes the information of physiographic indicators (elevation, latitude and longitude) and monthly mean fields from the satellite products: Tropical Rainfall Measuring Mission 2B31 microwave precipitation estimates, CMORPH microwave-plus-infrared based precipi-

tation estimates, monthly mean geostationary infrared brightness temperatures, and land surface temperature estimates. CHPclim uses a moving window regression for each grid cell with the latitude, longitude, and additional predictors from the satellite fields, elevation and slope. Secondly, the CHIRPS uses the Tropical Rainfall Measuring Mission Multi-satellite Precipitation Analysis version 7 (TMPA 3B42 v7) to calibrate global Cold Cloud Duration (CCD) rainfall. CDD measures the amount of time a given pixel has been covered by high cold cloud. Each month, the regression slopes and intercepts are derived using pentadal TMPA and TIR CCD data. These monthly 0.25° slopes and intercepts are resampled to a 0.05° grid, and used to produce 1981-present pentad precipitation estimates. CHIRP uses the 1981–2008 Globally Gridded Satellite (GriSat) archive produced by NOAA's National Climate Data Center and the 2000-present NOAA Climate Prediction Center dataset (CPC TIR). Finally, the CHIRPS station processing incorporates data from data streams and several private archives, as observations provided by national meteorological agencies. This station archive is used to define a set of global anchor station locations, to produce a more robust long-term time series. The CHIRPS station blending procedure is a modified inverse distance weighting algorithm, for any given pixel.

iii. We included now some additional references and links for the interested reader.

*) i. page 5, line 20. Bimonthly means "every 2 months". I guess you mean something like "semimonthly" (at least that is what the Merriam-Webster online dictionary suggests).

ii. GIMMS NDVI 3g uses the term bimonthly to refer to the temporal resolution of twice a month. However, the term "semimonthly" is now used instead of bimonthly in the manuscript to avoid the confusion with the definition for a temporal resolution of every two months.

iii. We included this information in the manuscript.

*) i. page 5, lines 27-30. Here you should give a some more detail on the validation of

processed NDVI values. Simple questions like: does it reproduce the seasonal cycle? are worth looking into and they give the reader the confidence that this might actually work! For example, a scatter plot of 15-day NDVI values against yearday for a few selected pixels.

ii. The reviewers suggested an improvement of the NDVI validation by showing a scatterplot of the day of the year and the 15-days NDVI (see Figure R2). In the manuscript a similar analysis is shown in Fig. 2b, by representing a boxplot of the mean monthly NDVI from July to June at the 23 studied locations. The figures R2 and 2b show the seasonality of the growing season that starts in September-October and ends in April-May.

iii. We consider that Fig. 2b gives consistent information of the NDVI dataset and there is no need to include Fig. R2 in the new manuscript. However, we improved the description of the NDVI behavior in relation with crop phenology in the new text.

*) i. page 6, line 18. When a correlation exceeding 0.7 is found, I take you label the satellite data as reliable, right? What if this threshold is not reached, what part of the analysis is then not possible?

ii. This study compared gauged and satellite precipitation data at 23 locations, from these 23 locations 21 (91%) had a significant correlation higher than 0.7 and two locations Charazani [6] and Colcha K [7] presented a significant correlation of 0.65. Previous studies have noted that correlations higher than 0.7 are reliable. If the correlation coefficient is small it would mean that the strength of the relationship is small, hence some regions have higher uncertainties for developing analysis. However, this study also shows the importance of the use of satellite data to develop climatological and hydrological analysis where gauged data are not available. We showed that satellite data could improve the temporal and spatial coverage of data. For instance, the present study is applicable for the entire Altiplano, considering also the areas where precipitation is not gauged. It is also important to consider the uncertainties of using satellite

data products due to measurement errors. Therefore, the authors suggest to use a databased combining gauged and satellite data.

iii. This explanation is now included in the manuscript.

*) i. page 6, line 31. Accumulated precipitation and accumulated temperature. Accumulated over what periods? In section 4.2 I read that precipitation is accumulated for 12 months, starting in July. Why is that? Also the start of the accumulation period is relevant, especially for the degree day metric.

ii. As described previously, the accumulation period is from July to June because this is the rain year period. And the study analyzed rainfed crop yields. In addition to the precipitation cycle, the precipitation shows also an inter-annual oscillation, showing the lowest temperatures occur during June and July, and the highest temperature in December. Therefore, the accumulated degree-day period starts in July and ends in June. As well, this period encloses the growing period of quinoa (September-April) and potato (October- March).

iii. The manuscript now describes the motivations for the accumulated period selection.

*) i. page 9, line 4-5. The relative error in the wet season is small because the amount of precipitation is so large - not because of small errors in the estimate. Please use a different metric to quantify the accuracy of the satellite-based precipitation.

ii. The CHIRPS satellite data product was validated using gauged precipitation at the 23 locations. The results show that the relative error is small for December, January, and February (the summer months). In agreement, Figure R3 shows the scatterplot of the monthly gauged and the satellite precipitation data, where the correlation has a median of 0.7*** (spearman correlation) for JFM, and the median correlation is 0.6*** for JJA at the 23 studied locations.

iii. The authors consider that the method used for the satellite validation analysis with a statistical approach is suitable to qualify the accuracy of the estimations. The authors

maintained the validation method, where it is computed: the spearman rank correlation, mean error, root mean squared error, and Nash Sutcliffe efficiency. However, the authors improved the description of the obtained results in the manuscript.

*) i. section 4.1 ends with the conclusions that CHIRPS is doing quite well, but the spatial aspect is lacking in this analysis. Are there stations/regions that are over or under estimated? A map with average RMSE values for January and July would be great here. Note that the distribution of rain gauges is not too homogeneous - so a little caution is in place here.

ii. RMSE and relative RMSE were used to evaluate CHIRPS simulations with gauged precipitation data. Figure R4 show the results of RMSE and relative RMSE for January and July. Considering that January is frequently the wettest month, the RMSE values are larger than other months. Most of the stations present a RMSE under 50 mm/month, except Charazani [6] and Copacabana [10] with a RMSE of 100 and 70 mm/month, respectively.

iii. The reviewer suggested to evaluate if the CHIRPS overestimate or underestimate gauged precipitation. This evaluation is described in the new manuscript. Figure R5 shows a map of the Altiplano with RMSE and ME values at the 23 studied locations for the period July 1981 to June 2016. New text has been included: "Most of the stations present a bias of +/-10 mm/month, meaning that the satellite simulations generally under- or overestimate 10 mm in relation to the ground precipitation data. However Charazani [6] present a large bias of 40.7 mm/month. As well, most of the stations present a RMSE between 15 to 30 mm/month, Charazani [6] had a RMSE of 58 mm/month". Figure R5 shows the locations where the satellite data underestimated (-10 to 0 mm/month) the gauged precipitation with white circles, and overestimated it with black circles (0.1 to 10 mm/month).

*) i. page 10, line 3. Here we see 'mean of maximum annual NDVI' - which is new to me (at least, I did not see it discussed earlier in the article).

ii. The analysis of NDVI and the crop yield relationship are described in the manuscript considering maximum annual NDVI (from July to June) of the crop land area (Fig. 1). The mean of the calculated maximum NDVI was computed in order to compare the results with annual crop yield for each region (La Paz, Oruro and Potosi).

To improve the analysis, a new computation was done considering months for which the phenology of quinoa and potato shows a main development (March to May). The maximum NDVI during these months was selected, and each NDVI grid was compared with the annual crop yield of quinoa and potato. The results of this analysis present a positive relationship between crop yield and vegetation. Only the pixels that presented a significant correlation above 0.6 were selected.

iii. The results of this analysis is described in the manuscript (please see Changes in manuscript).

*) i. page 11, line 23-25. Here the text seems to suggest that the 'ENSO warm phase' and 'extreme drought' are two separate things. Obviously, the effects of ENSO is the drought. Please rephrase.

ii. ENSO warm phase is associated with less precipitation, therefore ENSO could trigger of a drought event. However, precipitation variability does not only depend of ENSO, therefore in the text they are described separately.

iii. We included a discussion in the text.

*) i. fig. A3. Strange that winter values of relative ME and relative RMSE have wide distributions, while the individual months have very narrow distributions. Also: simply leaving-out values (like the monthly June and July values) when they do not fit in the plot, is not done. Include these in the plot and adjust the axis (put in a break - meaning skip some values in the axis - or use a non-linear axis).

ii. Winter months in the south hemisphere are June, July, and August, the relative RMSE and relative ME for the winter months present a wide distribution as well as a

seasonal distribution. These results are shown in Fig. A3 in the new manuscript.

iii. We confirmed that the results are coherent because the relative ME and relative RMSE show large distributions during winter season, and winter months (JJA), in contrast it shows small distributions for summer season and summer months (DJF). However, the description of the results was improved in the manuscript.

Changes in manuscript

In general, there was an improvement in the manuscript redaction and structure. The title was modified to "Drought risk assessment for the Bolivian Altiplano agriculture and its association with El Niño Southern Oscillation". The new objective of our paper is to present a methodology for assessing drought risk to mitigate its impacts on agriculture using observed gauge data and satellite based imagery data for the Bolivian Altiplano. The description of the modifications of the manuscript will be dived by sections: 1) Introduction, 2) Data Description, 3) Methods, 4) Results, 5) Discussion and, 6) Conclusion. The sections 1) Introduction and 2) Data Description where shortened to the relevant information. The section 3) Methods was modified as well, a new analysis of the relationship of the NDVI and the crop yield was done. The section with major changes is 4) Results, because we incorporated the new results of the relationship between NDVI and crop yield, and the new regression of NDVI and climate. Finally, the section 5) Discussion was also modified to suit with the new aim and findings, and 6) Conclusions" were adapted to the new manuscript content.

1. Introduction

Agricultural production is highly sensitive to weather extremes, including droughts and heat waves. Losses due to such extreme hazard events pose a significant challenge to farmers as well as governments worldwide (UNISDR, 2015). Worryingly, the scientific community predicts an amplification of these negative impacts due to future climate change (IPCC, 2013). Especially in developing countries as Bolivia, drought is as a major natural hazard. However, the impacts vary on a seasonal and annual timescale,

on the hazard intensity, and the capacity to prevent and respond to droughts (UNISDR, 2009). Regarding the former, the El Niño Southern Oscillation (ENSO) plays an important role. ENSO triggers droughts in several regions around the world, driving losses in agricultural crops and increased food insecurity (Kogan and Guo, 2017).

Bolivia have experienced large socio-economic losses in the past due to droughts. Generally, agricultural productivity in the Bolivian Altiplano is low due to high susceptibility to the climate, poor soil conditions, and the mainly manual labour. Poor agricultural production is also associated with the ENSO climate phenomena (Buxton et al., 2013). Droughts are generally driven by the ENSO warm phases (Vicente-Serrano et al., 2015; Garreaud and Aceituno, 2001; Thompson et al., 1984). Most important rainfed crops in the region include quinoa and potato. The Sustainable Development Goals (SDGs) state that the priority areas for adaptation to climate change are water and agriculture. These in turn, are related to the largest climate hazards including floods, droughts, and higher temperatures (UN, 2016). Additionally, the implementation of early warning is fundamental for drought disaster risk management, proactive planning, and mitigation policy measures in vulnerable regions, including Latin American countries such as Bolivia (Verbist et al., 2016).

Various studies of the relationship between ENSO and precipitation were developed previously, and they show a negative relationship between ENSO warm phase (El Nino) and precipitation, meaning that El Nino periods have been linked with a decrease of precipitation (Vicente-Serrano et al., 2015; Garreaud and Aceituno, 2001; Thompson et al., 1984; Francou et al., 2004; Vuille et al., 2000; Vuille, 1999). Other authors have studied ENSO-based seasonal forecasting (Lupo et al., 2017), and crop relation with climate variability (Garcia et al., 2007; Porter and Semenov, 2005). On the other hand, previous research of the relationship between ENSO and crop production was developed at global scale (Iizumi et al., 2014) and in South America (Anderson et al., 2017). Moreover, previous studies have already proposed ENSO based crop management in South America (Ramirez-Rodrigues et al., 2014). However, few studies have incorporated remote sensing data to relate drought events and ENSO (e.g. Liu and Juárez, 2001), and they are not developed in the Altiplano. In consequence, there is a gap of a drought risk assessment for the Bolivian Altiplano agriculture and its association with El Niño Southern Oscillation, using observed gauge data and satellite based imagery data. To lessen the long term impacts of these events, the national government has allocated a large budget for emergency operations to compensate part of the losses, which are usually evaluated in an ex-post approach. However, based on ENSO forecasting, an El Nino event can be predicted 1 to 7 months ahead (Tippett et al., 2012). For this time period it may be possible to implement ex-ante policies to reduce societal vulnerability to droughts, stressing preparedness, and improve risk management strategies. We are especially interested in how a risk based approach can be used to determine the potential need of resources as well as ways to determine hotspots where it is likely that these resources need to be distributed. A constraint to study drought occurrence is the uneven and scarce distribution of weather and crop related ground data in the region. We therefore use precipitation and vegetation satellite data that present full coverage of the spatial distribution in the study area. We combine this information with gauged precipitation, temperature, and crop yield data to enhance the knowledge and provide consistent results for climate and vegetation variability. Based on these observations, the objective of our paper is to present a methodology for assessing drought risk to mitigate its impacts on agriculture using observed gauged data and satellite based imagery data for the Bolivian Altiplano.

Our paper is organized as follows. Given the importance of data limitations in our case study region, we first introduce in section 2 the recently available datasets employed for our analysis. In section 3 we discuss the methods employed to (i) test the validity of the new datasets in our analysis, (ii) show how the relationship between climate and vegetation data was estimated, and (iii) how ENSO was incorporated in our analysis. After this, section 4 presents results and our proposed framework. Finally, section 5 ends with a conclusion, ways forward, and outlook to the future.
**2. Data Description**

The Bolivian Altiplano covers about 150,000 km2. It contains more than 70% of the total Altiplano surface area, the remaining percentage is located in southern Peru and northern Chile. La Paz, Oruro, and Potosi are the major administrative regions in the Bolivian Altiplano. The Altiplano has a pronounced southwest-northeast gradient (200–900 mm year−1) in annual precipitation with the wet season occurring from November to March (Garreaud et al., 2003). Over 60% of the annual precipitation occur during the summer months (DJF) in association with the South American Monsoon (SAM) (see Fig. A1).

**2.1 Climate Variables: Precipitation and Temperature Data**

Time series of observed monthly precipitation at 23 locations and mean, maximum, and minimum temperature at 15 locations from July 1981 to June 2016 were obtained from the National Service of Meteorology and Hydrology (SENAMHI; see Appendix Table A1). Initially, the available precipitation data sets included 65 gauges but only 23 were used since they have less than 10% of missing data. The data gaps were filled with the mean monthly value of the whole dataset to provide full time series. Outliers were identified by comparing with neighbouring monthly data. The inter-annual temperature at the 15 locations varied considerably between summer (DJFM) and winter (JJAS), including a larger variance for the minimum temperature (Fig. A2a). Regions close to the Lake Titicaca present lower inter-annual variability (Copacabana [10], Fig. A2b). In contrast, Uyuni [22] presents larger inter-annual oscillations (Fig. A2c). The precipitation gauges have an uneven spatial distribution and are mainly concentrated in the northern Bolivian Altiplano. To improve the spatial coverage of rainfall data, monthly quasi-rainfall time series from satellite data were included in our study. The Climate Hazards Group InfraRed Precipitation with station data (CHIRPS) quasi-global rainfall dataset was used. CHIRPS presents a 0.05° resolution satellite imagery and is a quasi-global rainfall dataset from 1981 to the near present with a satellite resolution of 0.05° (Funk et al., 2015). The information about CHIRPS is described in

http://chg.geog.ucsb.edu/data/chirps/.

2.2 El Niño Southern Oscillation Data

The Oceanic Niño Index (ONI) is usually used to identify El Niño (warm) and La Niña (cool) years (http://www.cpc.ncep.noaa.gov/). ONI is the 3 month running mean of Extended Reconstructed Sea Surface Temperature (ERSST v5) anomalies in the El Niño 3.4 region. The El Niño 3.4 anomalies represent the average equatorial SSTs in the equatorial Pacific Ocean (5oN to 5oS latitude, and 120o to 170oW longitude). Five consecutive overlapping three month periods at or above +0.5°C anomaly represents warm events (El Niño), and at or below the -0.5 anomaly are cold (La Niña) events.

2.3 Crop Production and Vegetation Data

As indicated above, quinoa and potato are the main crops in the Bolivian Altiplano and still gaining importance. The quinoa growing season is from September to April and, for potato from October to March. Data for quinoa and potato yield were obtained from the National Institute of Statistics (INE) of Bolivia from July 1981 to June 2016 for La Paz, Oruro, and Potosi (Fig. 1). No crop yield data at local scale are available and this is a major limitation that needs to be addressed in the future. The annual datasets represent production (t) in relation of the area (ha) at regional level.

Additionally, the Normalized Difference Vegetation Index (NDVI) can be used to estimate the vegetation vigour (Ji and Peters, 2003) and crop phenology (Beck et al., 2006). NDVI was assembled from the Advanced Very High Resolution Radiometer (AVHRR) sensors by the Global Inventory Monitoring and Modelling System (GIMMS) at semi-monthly (15 days) time steps with a spatial resolution of 0.08áţŠ. NDVI 3g.v1 (third generation GIMMS NDVI from AVHRR sensors) data set spans from July 1981 to December 2015. Note, the NDVI is an index that presents a range of values from 0 to 1, bare soil values are closer to 0, while dense vegetation has values close to 1 (Holben, 1986). NDVI 3g.v1 GIMMS provides information to differentiate valid values from possible errors due to snow, cloud, and interpolation errors. These errors were

eliminated from the dataset and where replaced with the nearest neighbour value. In our study we used the NDVI to simulate crop production.

**3. Methods**

**3.1 NDVI simulation of crop yield**

The maximum 15-days NDVI of March, April and May for every year was identified. Only the time from March to May was considered because this period represents the maximum phenological development of quinoa and potato crops. The maximum NDVI of each grid was compared to the annual crop yield at La Paz, Oruro, and Potosi. The NDVI grids and crop yield correlations larger than 0.6 (spearman correlation, p = 0.001) were considered as adequate for crop yield estimation, and only these grids were considered for further study. A similar approach was used by (Huang et al., 2014). Afterwards, a regression of the selected NDVI grids and the precipitation was developed. The satellite precipitation data was used for the regression. For this analysis, the NDVI grids were compared to the same spatial location of satellite precipitation data.

**4. Results**

**4.1 Validation of Chirps satellite precipitation data**

The mean annual gauged precipitation and CHIRPS satellite data product (Fig. R1) shows the relevance of the application of satellite data in the studied region.

The RMSE and ME (Fig. R5) shows the locations where satellite data overestimates or underestimates the gauged precipitation. Generally, the precipitation is under/overestimated in a range of -10 to +10 mm per month [6] (Fig. R5b). However Charazani [6] present a large bias of 40.7 mm/month. As well, most of the stations present a RMSE between 15 to 30 mm/month, Charazani [6] had a RMSE of 58 mm/month

**4.2 NDVI simulation of crop yield**

[Figure]

Figure R6 presents the locations where NDVI simulates the crop yield with a correlation larger than 0.6 (spearman correlation, p = 0.001). The NDVI grids that better simulates the quinoa crop are shown in Fig. R6a, and the NDVI that better simulates the potato crop are shown in Fig. R6b. We can see that NDVI can simulate the proper production of the crop land area in La Paz, Oruro, and Potosi. 4.3 NDVI association with precipitation

Only the NDVI grids with larger correlation of 0.6 with crop yield were considered for the climate regression. The results for stepwise linear regression between NDVI and accumulated precipitation were statistically significant at the 0.01 level. The analysis was firstly applied with the NDVI that best simulates the quinoa yield and afterwards with the NDVI that best simulates the potato yield. In La Paz, the mean correlation coefficient resulting from the regression of the NDVI that best simulates the quinoa and accumulated precipitation is 0.7 in La Paz, and above 0.6 in Oruro and Potosi. The regression of the NDVI that best estimates the potato yield and the accumulated precipitation shows a mean correlation coefficient of 0.7 in La Paz, 0.6 in Oruro, and 0.5 in Potosi.

Fig R6. Correlation coefficient (R2) of the regression of NDVI as the predictand, and precipitation as the predictor for the grids where NDVI better estimate the (a) quinoa and (b) potato yield. 4.4 Analysis of the ENSO impact on crop yield

Large reductions of crop yield are shown during El Nino events. The crop yield difference between the mean of the crop yield from 1981 to 2016 and the crop yield during a strong El Nino event 1982/1883 is shown in Fig R7. Quinoa yield reduces 65%, 10% and 73% at La Paz, Oruro, and Potosi respectively. And potato yield reduces 42%, 83% and 32% at La Paz, Oruro, and Potosi. In addition, the comparison between the mean NDVI from 1981 to 2016 and the NDVI during El Nino event of 1982-1983 is shown in Fig R8. Around the Lake Titicaca in the northern Altiplano the NDVI reduction is from 0 to 10%. And, in some locations of the southern Altiplano the reductions of NDVI are above 30%, and few locations reaches reductions above 40%, meaning great losses.

Fig. R7. Comparison of the crop production between the mean crop yield from 1981 to 2016 and the crop yield of 1982/1983 of a) quinoa and b) potato crops.

Fig. R8 NDVI reduction between the NDVI during a strong El Nino 1982-1983 and the mean NDVI from 1981 to 2016 for a) quinoa and b) potato crop.

4.5 Discussion

We employed a satellite product dataset and tested it for accuracy as well as performance to similar (but with coarser resolution) datasets available for our region. Using this dataset, it was shown that during El Nino years the crop yield reduces (Fig. R7), and as a consequence the socio-economic vulnerability of the farmers increases. Furthermore, it was found that NDVI can be related to crop yield and therefore, NDVI could be used to target specific hot spots depending on NDVIs availability at a local scale. As a consequence, ENSO forecasts as well as possible magnitudes of crop deficits could be established by the authorities, including identification of possible hotspots of crop deficits during the growing season. Our approach therefore, can not only help for determining the magnitude of assistance needed for farmers at the local level but also enable a pro-active approach to disaster risk management against droughts. This may include not only economic related instruments such as insurance but also risk reduction instruments such as irrigation needs. In fact, early warning based financing is gaining increasing attraction in some real world settings as it has several advantages. However, it should be acknowledged that large challenges still remains (French and Mechler, 2017). Drought severity could be measured via shifts from normal conditions of climatic parameters such as precipitation. As in our case, we not only provided shifts but the difference in risk for El Nino and neutral/moderate years. However, one of the main challenges of drought risk analysis is data-scarcity, e.g., low density or not evenly distributed stations for hydro-meteorological data networks, poor data quality due to missing data, and restricted use of data between government agencies or other institutions. As it was shown here, ENSO warm phase related characteristics are especially important in the context of extreme drought events and should therefore be

incorporated within early warning systems as standard practice. Despite these challenges for development of drought risk assessment, applications have been successful in the past. There are numerous cases in many countries around the globe. As in our case, particularly in the mid-latitudes weather patterns are strongly influenced by ENSO. Monitoring and predicting ENSO can therefore significantly contribute to reduce the risk of disasters.

4.6 Conclusions

This study is a first attempt to provide an agricultural drought risk assessment in relation to the ENSO phenomenon for the Bolivian Altiplano. Given the large differences in risk, and corresponding strategies to lessen the impacts could be implemented in the Bolivian Altiplano. In doing so, we introduced and tested a satellite product that was used for estimating crop risk. The ENSO impact on crop production was evaluated by studying the relation of crop yield, vegetation and climate variables, considering that El Nino generally drives a drought event. Our study provides valuable information for drought risk reduction, primarily by providing information of the hotspots where crop yield is more affected for droughts. Moreover, we showed that ENSO phases are strongly related to crop yield, and with this information the prediction of ENSO could be used to define risks in terms of decrease in crop yields in the studied region. While overall good fit among climate, ENSO, and crop yield variables were found, it is important to consider other parameters, such as evapotranspiration and soil moisture in improved models. With such information also agricultural models could be set up and risk management plans with better accuracy determined.

References

Anderson, W., Seager, R., Baethgen, W., and Cane, M.: Life cycles of agriculturally relevant ENSO teleconnections in North and South America, Int. J. Climatol., 37, 3297-3318, doi:10.1002/joc.4916, 2017.

Beck, P. S. A., Atzberger, C., Høgda, K. A., Johansen, B., and Skidmore,

A. K.: Improved monitoring of vegetation dynamics at very high latitudes: A new method using MODIS NDVI, Remote Sens. Environ., 100, 321-334, https://doi.org/10.1016/j.rse.2005.10.021, 2006.

Buxton, N., Escobar, M., Purkey, D., and Lima, N.: Water scarcity, climate change and Bolivia: Planning for climate uncertainties, Stockholm Environment Institute U.S. Center – Davis Office Davis, USA, 4, 2013.

Duan, Y., Wilson, A. M., and Barros, A. P.: Scoping a field experiment: error diagnostics of TRMM precipitation radar estimates in complex terrain as a basis for IPHEx2014, Hydrol. Earth Syst. Sci., 19, 1501-1520, 10.5194/hess-19-1501-2015, 2015.

Francou, B., Vuille, M., Favier, V., and Cáceres, B.: New evidence for an ENSO impact on low-latitude glaciers: Antizana 15, Andes of Ecuador, 0°28′S, J. Geophys. Res. Atmos., 109, doi:10.1029/2003JD004484, 2004.

French, A., and Mechler, R.: Managing El Niño Risks Under Uncertainty in Peru: Learning from the past for a more disaster-resilient future, International Institute for Applied Systems Analysis, Laxenburg, Austria, 2017.

Funk, C., Peterson, P., Landsfeld, M., Pedreros, D., Verdin, J., Shukla, S., Husak, G., Rowland, J., Harrison, L., Hoell, A., and Michaelsen, J.: The climate hazards infrared precipitation with stations—a new environmental record for monitoring extremes, A Nature Research Journal, 2, 150066, 10.1038/sdata.2015.66, 2015.

Garcia, M., Raes, D., Jacobsen, S. E., and Michel, T.: Agroclimatic constraints for rainfed agriculture in the Bolivian Altiplano, J Arid Environ., 71, 109-121, https://doi.org/10.1016/j.jaridenv.2007.02.005, 2007.

Garreaud, R., Vuille, M., and Clement, A. C.: The climate of the Altiplano: observed current conditions and mechanisms of past changes, Palaeogeogr. Palaeoclimatol. Palaeoecol., 194, 5-22, 10.1016/S0031-0182(03)00269-4, 2003.

Garreaud, R. D., and Aceituno, P.: Interannual rainfall variability over

the South American Altiplano, J. Clim., 14, 2779-2789, 10.1175/1520-0442(2001)014<2779:Irvots>2.0.Co;2, 2001. Garreaud, R. D.: The Andes climate and weather, Adv. Geosci., 22, 3-11, 10.5194/adgeo-22-3-2009, 2009.

Holben, B. N.: Characteristics of maximum-value composite images from temporal AVHRR data, Int J Remote Sens, 7, 1417-1434, 10.1080/01431168608948945, 1986.

Huang, J., Wang, H., Dai, Q., and Han, D.: Analysis of NDVI Data for Crop Identification and Yield Estimation, IEEE Journal of Selected Topics in Applied Earth Observations and Remote Sensing, 7, 4374-4384, 10.1109/JSTARS.2014.2334332, 2014.

Iizumi, T., Luo, J.-J., Challinor, A. J., Sakurai, G., Yokozawa, M., Sakuma, H., Brown, M. E., and Yamagata, T.: Impacts of El Niño Southern Oscillation on the global yields of major crops, Nature Communications, 5, 3712, 10.1038/ncomms4712 https://www.nature.com/articles/ncomms4712#supplementary-information, 2014.

IPCC: Climate Change 2013: The Physical Science Basis. Contribution of Working Group I to the Fifth Assessment Report of the Intergovernmental Panel on Climate Change, edited by: Stocker, T. F., Qin, D., Plattner, G.-K., Tignor, M., Allen, S. K., Boschung, J., Nauels, A., Xia, Y., Bex, V., and Midgley, G. F., Cambridge University Press, Cambridge, United Kingdom and New York, NY, USA, 1535 pp., 2013. Ji, L., and Peters, A. J.: Assessing vegetation response to drought in the northern Great Plains using vegetation and drought indices, Remote Sens. Environ., 87, 85-98, https://doi.org/10.1016/S0034-4257(03)00174-3, 2003.

Kogan, F., and Guo, W.: Strong 2015–2016 El Niño and implication to global ecosystems from space data, Int J Remote Sens, 38, 161-178, 10.1080/01431161.2016.1259679, 2017.

Liu, W. T., and Juárez, R. I. N.: ENSO drought onset prediction in northeast Brazil using NDVI, Int J Remote Sens, 22, 3483-3501, 10.1080/01431160010006430, 2001.

Lupo, A. R., Garcia, M., Rojas, K., and Gilles, J.: ENSO Related Seasonal Range

Prediction over South America, Proceedings, 1, 682, 2017.

Nash, J. E., and Sutcliffe, J. V.: River flow forecasting through conceptual models part I — A discussion of principles, J. Hydrol., 10, 282-290, https://doi.org/10.1016/0022-1694(70)90255-6, 1970.

Porter, J. R., and Semenov, M. A.: Crop responses to climatic variation, Philosophical Transactions of the Royal Society B: Biological Sciences, 360, 2021-2035, 10.1098/rstb.2005.1752, 2005.

Ramirez-Rodrigues, M. A., Asseng, S., Fraisse, C., Stefanova, L., and Eisenkolbi, A.: Tailoring wheat management to ENSO phases for increased wheat production in Paraguay, Climate Risk Management, 3, 24-38, https://doi.org/10.1016/j.crm.2014.06.001, 2014.

Seiler, C., Hutjes, R. W. A., and Kabat, P.: Climate Variability and Trends in Bolivia, J. Appl. Meteorol. Clim., 52, 130-146, 10.1175/Jamc-D-12-0105.1, 2013.

Thompson, L. G., Mosley-Thompson, E., and Arnao, B. M.: El Nino-Southern Oscillation events recorded in the stratigraphy of the tropical Quelccaya ice cap, Peru, Science, 226, 50-53, 10.1126/science.226.4670.50, 1984.

Tippett, M. K., Barnston, A. G., and Li, S.: Performance of Recent Multimodel ENSO Forecasts, J. Appl. Meteorol. Clim., 51, 637-654, 10.1175/jamc-d-11-093.1, 2012.

UN: The Sustainable Development Goals Report 2016, United Nations New York, USA, 2016. UNISDR: Drought Risk Reduction Framework and Practices: Contributing to the Implementation of the Hyogo Framework for Action, United Nations secretariat of the International Strategy for Disaster Reduction (UNISDR), Geneva, Switzerland, 2009.

UNISDR: Making Development Sustainable: The Future of Disaster Risk Management. Global Assessment Report on Disaster Risk Reduction, United Nations Office for Disaster Risk Reduction (UNISDR), Geneva, Switzerland, 266, 2015.

[Figure]

Verbist, K., Amani, A., Mishra, A., and Cisneros, B. J.: Strengthening drought risk management and policy: UNESCO International Hydrological Programme's case studies from Africa and Latin America and the Caribbean, Water Policy, 18, 245-261, 10.2166/wp.2016.223, 2016.

Vicente-Serrano, S. M., Chura, O., López-Moreno, J. I., Azorin-Molina, C., Sanchez-Lorenzo, A., Aguilar, E., Moran-Tejeda, E., Trujillo, F., Martínez, R., and Nieto, J. J.: Spatio-temporal variability of droughts in Bolivia: 1955–2012, Int. J. Climatol., 35, 3024-3040, 10.1002/joc.4190, 2015.

Vuille, M.: Atmospheric circulation over the Bolivian Altiplano during dry and wet periods and extreme phases of the Southern Oscillation, Int. J. Climatol., 19, 1579-1600, 10.1002/(SICI)1097-0088(19991130)19:14<1579::AID-JOC441>3.0.CO;2-N, 1999.

Vuille, M., Bradley, R. S., and Keimig, F.: Interannual climate variability in the Central Andes and its relation to tropical Pacific and Atlantic forcing, J. Geophys. Res. Atmos., 105, 12447-12460, 10.1029/2000JD900134, 2000.

Wilks, D. S.: Statistical Methods in the Atmospheric Sciences, second ed., Academic Press, 2006.

Please also note the supplement to this comment:
https://www.nat-hazards-earth-syst-sci-discuss.net/nhess-2018-133/nhess-2018-133-AC2-supplement.pdf

---

## Author Comment (AC3) · 24 Aug 2018

Response to reviewers

The authors thank the editor, reviewers, and third person's comments for time spent and efforts to improve the manuscript. We have revised the manuscript accordingly taking the reviewers' and third person's comments into due consideration. Below follow answers to the reviewer's comments and description of actions taken.

While the reviewers indicated that the article is of substantial interest and relevant for the journal they criticized the misleading title as well as the analysis to be "poorly executed". Furthermore, they reviewers indicated that the quality of the presented material is "not satisfactory" to support the results found. We agree with the reviewers in the sense that one major restriction to provide a sophisticated model between satellite imagery and agriculture risk is data limitation. However, the unavailability of data to be used for a sophisticated drought modelling approach are very common in a developing country context (World Bank 2016). One way to overcome this challenge is to apply a so-called iterative risk management approach, e.g. starting with baseline estimates using the best data currently available and updating risk estimates continuously over time (see IPCC 2012). Currently there are no studies for the Altiplano which relates satellite sourced imagery with agriculture risk, including possible associations with ENSO. The article tries to fill this gap. We acknowledge the fact that there are other studies in other countries which are using more sophisticated models, however, the situation in the Altiplano, especially the data limitations, restricts the use of such models and we provide a way forward to improve the data situation using high-resolution satellite imagery with a probabilistic approach for agriculture risk. Hence, for the current situation in Bolivia our approach can be regarded as one way forward, and can be used as a baseline case for further analysis in the future. As indicated, the situation is quite similar in other developing countries around the world and our approach can be seen as one way forward how to implement drought risk management under data scarcity including the important connection with the ENSO phenomenon. We also provided now much more detail on the strengths and limitations of the approach, including a detailed uncertainty analysis. Please find our detailed responses to the reviewer comments below.

The response to reviewers are structured following the recommendations of the editors:

i. Comment from referee

ii. Authors' response

iii. Author's changes in manuscript.

In addition the last section "Changes in manuscript" describes in more detail the author's changes in manuscript.

Short comment: V. Moya Quiroga 1. i. The term early warning usually refers to a time anticipation; X time before a given event the warning is emitted. In the present case, readers would expect to see the NDVI or ENSO conditions that represent a high probability of drought some months before the drought. For instance, if the September ENSO is X, there is a Y probability of drought. However, the article doesn't show any early warning. The article only analyzes the some ENSO relationships, which is nothing new. Several previous researchers (not referenced in the literature review) have already analyzed the relationship between ENSO and crop production in South America (Izumi et al., 2014; Anderson et al., 2016). Moreover, previous studies have already proposed ENSO based crop management policies in South America (Ramirez-Rodrigues et al., 2014)

ii. The authors agree with the comment, and we changed the title and aim to the study. The new title is "Drought risk assessment for the Bolivian Altiplano agriculture and its association with El Niño Southern Oscillation". And the new aim is to present a methodology for assessing drought risk to mitigate its impacts on agriculture using observed gauge data and satellite based imagery data for the Bolivian Altiplano.

In contrast of the mentioned previous research of the ENSO-based relationship with crop production and climate variables, this research aims to downscale the regional information at a local level, this permits to define hotspots where droughts could affect in large extend the agricultural production, and this information can be useful for risk management.

iii. The authors improved the introduction section and included the findings of previous research related with the study. 2. i. It is not clear which satellite data was used. The title suggests "high resolution", but the manuscript mentions 0.05 degree (about 5 km). The term high resolution satellite imagery usually refers to less than 90 m or 50 m resolution.

ii. NDVI has a grid of 0.08 degree and CHIPRS satellite precipitation data has 0.05 degree of spatial resolution.

iii. The manuscript was modified and it does not include the term "high resolution" to avoid confusions.

3. i. The manuscript only shows the results of 1 correlation for 1 station (NDVI correlation for Tihuanacu), which is supposed to be the one with the highest correlation (Pg9 Ln21). This brings several doubts. *How is the correlation in other locations? (Maybe a thematic correlation map would be more appropriate) *Even though Tihuanacu has the highest correlation, the correlation is still quite weak (0.41). Does it means that the other correlations were even lower? *Manuscript states that it considered 23 stations, most them located in La Paz. But then it presents an average value for each department. This procedure is not appropriate for several reasons. For instance: a) It is not fair to compare the average of La Paz (more than 15 stations) with the average of Potosi (only 3 stations poorly distributed) b) In the present methodology the stations were clustered based on political boundaries. Does it mean that bolivian climate behaves according to its political boundaries? I don't believe so. For instance, the stations near Titikaka lake (in La Paz) have very different climatic characteristics than other stations also in La Paz (for instance El Alto or Santiago de Machaca).

ii. A new analysis was done in order to find out the relationship between NDVI and crop production. The maximum NDVI during the months of major development of potato and quinoa crops (March to May) was selected and compared with the annual crop yield of quinoa and potato at La Paz, Oruro and Potosi. The results of this analysis present a positive relationship between crop yield and vegetation. Only the pixels that presented a significant correlation above 0.6 were selected for the following regression with the climate variables.

Using the new described methodology we avoided possible errors that could bring the analysis of the mean NDVI and the annual crop at regional level. As the comment

mentions, the Bolivian climate does not behave as the political boundary, and therefore the mean of the maximum NDVI at the administrative level is no longer used in the manuscript. * i. The correlations from FigA6 are different than the values from Table 2 (because of rounding decimals).

ii. The authors revised and corrected the decimals shown in the figures and tables.

* i. Manuscript states that Tihuanacu has the highest correlation, but Table 2 shows that Potosi has higher correlation.

ii. Manuscript states that Tihuanacu has the highest correlation coefficient between NDVI and climate variables, and Table 2 shows that Potosi has higher correlation (spearman correlation, p=0.001) between crop yield and NDVI. These are two different analysis, and therefore the results are also different.

iii. We applied a different analysis to seek the relationship of NDVI and crop production. We no longer considered the relationship of the mean NDVI and crop yield at local level (La Paz, Oruro, and Potosi), instead the analysis the relationship of every NDVI grid (0.08áȚŠ) with the crop yield. The new results are included in the manuscript.

* i. The highest correlation (0.51) is still a weak correlation. What is the confidence of a prediction based on such a weak linear correlation? What would be the confidence of predictions for Oruro based on a linear correlation of 0.12? Unfortunately, it seems that the present results suggest that the present methodology does not provide a reliable drought estimations.

ii. The authors agree with the comment and therefore we improved the analysis by selecting the locations where NDVI and crop production have a correlation above 0.6*** for the subsequent regression of NDVI and the climate variables.

iii. The method and results of the analysis are described in Changes in manuscript

Changes in manuscript

In general, there was an improvement in the manuscript redaction and structure. The title was modified to "Drought risk assessment for the Bolivian Altiplano agriculture and its association with El Niño Southern Oscillation". The new objective of our paper is to present a methodology for assessing drought risk to mitigate its impacts on agriculture using observed gauge data and satellite based imagery data for the Bolivian Altiplano. The description of the modifications of the manuscript will be dived by sections: 1) Introduction, 2) Data Description, 3) Methods, 4) Results, 5) Discussion and, 6) Conclusion. The sections 1) Introduction and 2) Data Description where shortened to the relevant information. The section 3) Methods was modified as well, a new analysis of the relationship of the NDVI and the crop yield was done. The section with major changes is 4) Results, because we incorporated the new results of the relationship between NDVI and crop yield, and the new regression of NDVI and climate. Finally, the section 5) Discussion was also modified to suit with the new aim and findings, and 6) Conclusions" were adapted to the new manuscript content.

1. Introduction

Agricultural production is highly sensitive to weather extremes, including droughts and heat waves. Losses due to such extreme hazard events pose a significant challenge to farmers as well as governments worldwide (UNISDR, 2015). Worryingly, the scientific community predicts an amplification of these negative impacts due to future climate change (IPCC, 2013). Especially in developing countries as Bolivia, drought is as a major natural hazard. However, the impacts vary on a seasonal and annual timescale, on the hazard intensity, and the capacity to prevent and respond to droughts (UNISDR, 2009). Regarding the former, the El Niño Southern Oscillation (ENSO) plays an important role. ENSO triggers droughts in several regions around the world, driving losses in agricultural crops and increased food insecurity (Kogan and Guo, 2017).

Bolivia have experienced large socio-economic losses in the past due to droughts. Generally, agricultural productivity in the Bolivian Altiplano is low due to high susceptibility to the climate, poor soil conditions, and the mainly manual labour. Poor agricultural production is also associated with the ENSO climate phenomena (Buxton et al., 2013). Droughts are generally driven by the ENSO warm phases (Vicente-Serrano et al., 2015; Garreaud and Aceituno, 2001; Thompson et al., 1984). Most important rainfed crops in the region include quinoa and potato. The Sustainable Development Goals (SDGs) state that the priority areas for adaptation to climate change are water and agriculture. These in turn, are related to the largest climate hazards including floods, droughts, and higher temperatures (UN, 2016). Additionally, the implementation of early warning is fundamental for drought disaster risk management, proactive planning, and mitigation policy measures in vulnerable regions, including Latin American countries such as Bolivia (Verbist et al., 2016).

Various studies of the relationship between ENSO and precipitation were developed previously, and they show a negative relationship between ENSO warm phase (El Nino) and precipitation, meaning that El Nino periods have been linked with a decrease of precipitation (Vicente-Serrano et al., 2015; Garreaud and Aceituno, 2001; Thompson et al., 1984; Francou et al., 2004; Vuille et al., 2000; Vuille, 1999). Other authors have studied ENSO-based seasonal forecasting (Lupo et al., 2017), and crop relation with climate variability (Garcia et al., 2007; Porter and Semenov, 2005). On the other hand, previous research of the relationship between ENSO and crop production was developed at global scale (Iizumi et al., 2014) and in South America (Anderson et al., 2017). Moreover, previous studies have already proposed ENSO based crop management in South America (Ramirez-Rodrigues et al., 2014). However, few studies have incorporated remote sensing data to relate drought events and ENSO (e.g. Liu and Juárez, 2001), and they are not developed in the Altiplano. In consequence, there is a gap of a drought risk assessment for the Bolivian Altiplano agriculture and its association with El Niño Southern Oscillation, using observed gauge data and satellite based imagery data. To lessen the long term impacts of these events, the national government has allocated a large budget for emergency operations to compensate part of the losses, which are usually evaluated in an ex-post approach. However, based on ENSO forecasting, an El Nino event can be predicted 1 to 7 months ahead (Tippett et al., 2012).
[Figure]

For this time period it may be possible to implement ex-ante policies to reduce societal vulnerability to droughts, stressing preparedness, and improve risk management strategies. We are especially interested in how a risk based approach can be used to determine the potential need of resources as well as ways to determine hotspots where it is likely that these resources need to be distributed. A constraint to study drought occurrence is the uneven and scarce distribution of weather and crop related ground data in the region. We therefore use precipitation and vegetation satellite data that present full coverage of the spatial distribution in the study area. We combine this information with gauged precipitation, temperature, and crop yield data to enhance the knowledge and provide consistent results for climate and vegetation variability. Based on these observations, the objective of our paper is to present a methodology for assessing drought risk to mitigate its impacts on agriculture using observed gauged data and satellite based imagery data for the Bolivian Altiplano.

Our paper is organized as follows. Given the importance of data limitations in our case study region, we first introduce in section 2 the recently available datasets employed for our analysis. In section 3 we discuss the methods employed to (i) test the validity of the new datasets in our analysis, (ii) show how the relationship between climate and vegetation data was estimated, and (iii) how ENSO was incorporated in our analysis. After this, section 4 presents results and our proposed framework. Finally, section 5 ends with a conclusion, ways forward, and outlook to the future.

2. Data Description

The Bolivian Altiplano covers about 150,000 km2. It contains more than 70% of the total Altiplano surface area, the remaining percentage is located in southern Peru and northern Chile. La Paz, Oruro, and Potosi are the major administrative regions in the Bolivian Altiplano. The Altiplano has a pronounced southwest-northeast gradient (200–900 mm year−1) in annual precipitation with the wet season occurring from November to March (Garreaud et al., 2003). Over 60% of the annual precipitation occur during the summer months (DJF) in association with the South American Monsoon (SAM) (see

[Figure]

Fig. A1).

**2.1 Climate Variables: Precipitation and Temperature Data**

Time series of observed monthly precipitation at 23 locations and mean, maximum, and minimum temperature at 15 locations from July 1981 to June 2016 were obtained from the National Service of Meteorology and Hydrology (SENAMHI; see Appendix Table A1). Initially, the available precipitation data sets included 65 gauges but only 23 were used since they have less than 10% of missing data. The data gaps were filled with the mean monthly value of the whole dataset to provide full time series. Outliers were identified by comparing with neighbouring monthly data. The inter-annual temperature at the 15 locations varied considerably between summer (DJFM) and winter (JJAS), including a larger variance for the minimum temperature (Fig. A2a). Regions close to the Lake Titicaca present lower inter-annual variability (Copacabana [10], Fig. A2b). In contrast, Uyuni [22] presents larger inter-annual oscillations (Fig. A2c). The precipitation gauges have an uneven spatial distribution and are mainly concentrated in the northern Bolivian Altiplano. To improve the spatial coverage of rainfall data, monthly quasi-rainfall time series from satellite data were included in our study. The Climate Hazards Group InfraRed Precipitation with station data (CHIRPS) quasi-global rainfall dataset was used. CHIRPS presents a $0.05°$ resolution satellite imagery and is a quasi-global rainfall dataset from 1981 to the near present with a satellite resolution of $0.05°$ (Funk et al., 2015). The information about CHIRPS is described in http://chg.geog.ucsb.edu/data/chirps/.

**2.2 El Niño Southern Oscillation Data**

The Oceanic Niño Index (ONI) is usually used to identify El Niño (warm) and La Niña (cool) years (http://www.cpc.ncep.noaa.gov/). ONI is the 3 month running mean of Extended Reconstructed Sea Surface Temperature (ERSST v5) anomalies in the El Niño 3.4 region. The El Niño 3.4 anomalies represent the average equatorial SSTs in the equatorial Pacific Ocean (5oN to 5oS latitude, and 120o to 170oW longitude). Five

consecutive overlapping three month periods at or above +0.5°C anomaly represents warm events (El Niño), and at or below the -0.5 anomaly are cold (La Niña) events.

2.3 Crop Production and Vegetation Data

As indicated above, quinoa and potato are the main crops in the Bolivian Altiplano and still gaining importance. The quinoa growing season is from September to April and, for potato from October to March. Data for quinoa and potato yield were obtained from the National Institute of Statistics (INE) of Bolivia from July 1981 to June 2016 for La Paz, Oruro, and Potosi (Fig. 1). No crop yield data at local scale are available and this is a major limitation that needs to be addressed in the future. The annual datasets represent production (t) in relation of the area (ha) at regional level.

Additionally, the Normalized Difference Vegetation Index (NDVI) can be used to estimate the vegetation vigour (Ji and Peters, 2003) and crop phenology (Beck et al., 2006). NDVI was assembled from the Advanced Very High Resolution Radiometer (AVHRR) sensors by the Global Inventory Monitoring and Modelling System (GIMMS) at semi-monthly (15 days) time steps with a spatial resolution of 0.08áţŠ. NDVI 3g.v1 (third generation GIMMS NDVI from AVHRR sensors) data set spans from July 1981 to December 2015. Note, the NDVI is an index that presents a range of values from 0 to 1, bare soil values are closer to 0, while dense vegetation has values close to 1 (Holben, 1986). NDVI 3g.v1 GIMMS provides information to differentiate valid values from possible errors due to snow, cloud, and interpolation errors. These errors were eliminated from the dataset and where replaced with the nearest neighbour value. In our study we used the NDVI to simulate crop production.

3. Methods

3.1 NDVI simulation of crop yield

The maximum 15-days NDVI of March, April and May for every year was identified. Only the time from March to May was considered because this period represents the

maximum phenological development of quinoa and potato crops. The maximum NDVI of each grid was compared to the annual crop yield at La Paz, Oruro, and Potosi. The NDVI grids and crop yield correlations larger than 0.6 (spearman correlation, p = 0.001) were considered as adequate for crop yield estimation, and only these grids were considered for further study. A similar approach was used by (Huang et al., 2014). Afterwards, a regression of the selected NDVI grids and the precipitation was developed. The satellite precipitation data was used for the regression. For this analysis, the NDVI grids were compared to the same spatial location of satellite precipitation data.

4. Results

4.1 Validation of Chirps satellite precipitation data

The mean annual gauged precipitation and CHIRPS satellite data product (Fig. R1) shows the relevance of the application of satellite data in the studied region.

The RMSE and ME (Fig. R5) shows the locations where satellite data overestimates or underestimates the gauged precipitation. Generally, the precipitation is under/overestimated in a range of -10 to +10 mm per month [6] (Fig. R5b). However Charazani [6] present a large bias of 40.7 mm/month. As well, most of the stations present a RMSE between 15 to 30 mm/month, Charazani [6] had a RMSE of 58 mm/month.

4.2 NDVI simulation of crop yield

Figure R6 presents the locations where NDVI simulates the crop yield with a correlation larger than 0.6 (spearman correlation, p = 0.001). The NDVI grids that better simulates the quinoa crop are shown in Fig. R6a, and the NDVI that better simulates the potato crop are shown in Fig. R6b. We can see that NDVI can simulate the proper production of the crop land area in La Paz, Oruro, and Potosi.

4.3 NDVI association with precipitation

Only the NDVI grids with larger correlation of 0.6 with crop yield were considered for

the climate regression. The results for stepwise linear regression between NDVI and accumulated precipitation were statistically significant at the 0.01 level. The analysis was firstly applied with the NDVI that best simulates the quinoa yield and afterwards with the NDVI that best simulates the potato yield. In La Paz, the mean correlation coefficient resulting from the regression of the NDVI that best simulates the quinoa and accumulated precipitation is 0.7 in La Paz, and above 0.6 in Oruro and Potosi. The regression of the NDVI that best estimates the potato yield and the accumulated precipitation shows a mean correlation coefficient of 0.7 in La Paz, 0.6 in Oruro, and 0.5 in Potosi.

4.4 Analysis of the ENSO impact on crop yield

Large reductions of crop yield are shown during El Nino events. The crop yield difference between the mean of the crop yield from 1981 to 2016 and the crop yield during a strong El Nino event 1982/1883 is shown in Fig R7. Quinoa yield reduces 65%, 10% and 73% at La Paz, Oruro, and Potosi respectively. And potato yield reduces 42%, 83% and 32% at La Paz, Oruro, and Potosi. In addition, the comparison between the mean NDVI from 1981 to 2016 and the NDVI during El Nino event of 1982-1983 is shown in Fig R8. Around the Lake Titicaca in the northern Altiplano the NDVI reduction is from 0 to 10%. And, in some locations of the southern Altiplano the reductions of NDVI are above 30%, and few locations reaches reductions above 40%, meaning great losses.

5 Discussion

We employed a satellite product dataset and tested it for accuracy as well as performance to similar (but with coarser resolution) datasets available for our region. Using this dataset, it was shown that during El Nino years the crop yield reduces (Fig. R7), and as a consequence the socio-economic vulnerability of the farmers increases. Furthermore, it was found that NDVI can be related to crop yield and therefore, NDVI could be used to target specific hot spots depending on NDVIs availability at a local scale. As a consequence, ENSO forecasts as well as possible magnitudes of crop deficits

could be established by the authorities, including identification of possible hotspots of crop deficits during the growing season. Our approach therefore, can not only help for determining the magnitude of assistance needed for farmers at the local level but also enable a pro-active approach to disaster risk management against droughts. This may include not only economic related instruments such as insurance but also risk reduction instruments such as irrigation needs. In fact, early warning based financing is gaining increasing attraction in some real world settings as it has several advantages. However, it should be acknowledged that large challenges still remains (French and Mechler, 2017). Drought severity could be measured via shifts from normal conditions of climatic parameters such as precipitation. As in our case, we not only provided shifts but the difference in risk for El Nino and neutral/moderate years. However, one of the main challenges of drought risk analysis is data-scarcity, e.g., low density or not evenly distributed stations for hydro-meteorological data networks, poor data quality due to missing data, and restricted use of data between government agencies or other institutions. As it was shown here, ENSO warm phase related characteristics are especially important in the context of extreme drought events and should therefore be incorporated within early warning systems as standard practice. Despite these challenges for development of drought risk assessment, applications have been successful in the past. There are numerous cases in many countries around the globe. As in our case, particularly in the mid-latitudes weather patterns are strongly influenced by ENSO. Monitoring and predicting ENSO can therefore significantly contribute to reduce the risk of disasters.

6 Conclusions

This study is a first attempt to provide an agricultural drought risk assessment in relation to the ENSO phenomenon for the Bolivian Altiplano. Given the large differences in risk, and corresponding strategies to lessen the impacts could be implemented in the Bolivian Altiplano. In doing so, we introduced and tested a satellite product that was used for estimating crop risk. The ENSO impact on crop production was evaluated

by studying the relation of crop yield, vegetation and climate variables, considering that El Nino generally drives a drought event. Our study provides valuable information for drought risk reduction, primarily by providing information of the hotspots where crop yield is more affected for droughts. Moreover, we showed that ENSO phases are strongly related to crop yield, and with this information the prediction of ENSO could be used to define risks in terms of decrease in crop yields in the studied region. While overall good fit among climate, ENSO, and crop yield variables were found, it is important to consider other parameters, such as evapotranspiration and soil moisture in improved models. With such information also agricultural models could be set up and risk management plans with better accuracy determined.

References

Anderson, W., Seager, R., Baethgen, W., and Cane, M.: Life cycles of agriculturally relevant ENSO teleconnections in North and South America, Int. J. Climatol., 37, 3297-3318, doi:10.1002/joc.4916, 2017.

Beck, P. S. A., Atzberger, C., Høgda, K. A., Johansen, B., and Skidmore, A. K.: Improved monitoring of vegetation dynamics at very high latitudes: A new method using MODIS NDVI, Remote Sens. Environ., 100, 321-334, https://doi.org/10.1016/j.rse.2005.10.021, 2006.

Buxton, N., Escobar, M., Purkey, D., and Lima, N.: Water scarcity, climate change and Bolivia: Planning for climate uncertainties, Stockholm Environment Institute U.S. Center – Davis Office Davis, USA, 4, 2013.

Duan, Y., Wilson, A. M., and Barros, A. P.: Scoping a field experiment: error diagnostics of TRMM precipitation radar estimates in complex terrain as a basis for IPHEx2014, Hydrol. Earth Syst. Sci., 19, 1501-1520, 10.5194/hess-19-1501-2015, 2015.

Francou, B., Vuille, M., Favier, V., and Cáceres, B.: New evidence for an ENSO impact on low-latitude glaciers: Antizana 15, Andes of Ecuador, 0°28′S, J. Geophys. Res.

[Figure]

Atmos., 109, doi:10.1029/2003JD004484, 2004.

French, A., and Mechler, R.: Managing El Niño Risks Under Uncertainty in Peru: Learning from the past for a more disaster-resilient future, International Institute for Applied Systems Analysis, Laxenburg, Austria, 2017.

Funk, C., Peterson, P., Landsfeld, M., Pedreros, D., Verdin, J., Shukla, S., Husak, G., Rowland, J., Harrison, L., Hoell, A., and Michaelsen, J.: The climate hazards infrared precipitation with stations—a new environmental record for monitoring extremes, A Nature Research Journal, 2, 150066, 10.1038/sdata.2015.66, 2015.

Garcia, M., Raes, D., Jacobsen, S. E., and Michel, T.: Agroclimatic constraints for rainfed agriculture in the Bolivian Altiplano, J Arid Environ., 71, 109-121, https://doi.org/10.1016/j.jaridenv.2007.02.005, 2007.

Garreaud, R., Vuille, M., and Clement, A. C.: The climate of the Altiplano: observed current conditions and mechanisms of past changes, Palaeogeogr. Palaeoclimatol. Palaeoecol., 194, 5-22, 10.1016/S0031-0182(03)00269-4, 2003.

Garreaud, R. D., and Aceituno, P.: Interannual rainfall variability over the South American Altiplano, J. Clim., 14, 2779-2789, 10.1175/1520-0442(2001)014<2779:Irvots>2.0.Co;2, 2001. Garreaud, R. D.: The Andes climate and weather, Adv. Geosci., 22, 3-11, 10.5194/adgeo-22-3-2009, 2009.

Holben, B. N.: Characteristics of maximum-value composite images from temporal AVHRR data, Int J Remote Sens, 7, 1417-1434, 10.1080/01431168608948945, 1986.

Huang, J., Wang, H., Dai, Q., and Han, D.: Analysis of NDVI Data for Crop Identification and Yield Estimation, IEEE Journal of Selected Topics in Applied Earth Observations and Remote Sensing, 7, 4374-4384, 10.1109/JSTARS.2014.2334332, 2014.

Iizumi, T., Luo, J.-J., Challinor, A. J., Sakurai, G., Yokozawa, M., Sakuma, H., Brown, M. E., and Yamagata, T.: Impacts of El Niño Southern Oscillation on the global yields of major crops, Nature Communications, 5, 3712, 10.1038/ncomms4712

https://www.nature.com/articles/ncomms4712#supplementary-information, 2014.

IPCC: Climate Change 2013: The Physical Science Basis. Contribution of Working Group I to the Fifth Assessment Report of the Intergovernmental Panel on Climate Change, edited by: Stocker, T. F., Qin, D., Plattner, G.-K., Tignor, M., Allen, S. K., Boschung, J., Nauels, A., Xia, Y., Bex, V., and Midgley, G. F., Cambridge University Press, Cambridge, United Kingdom and New York, NY, USA, 1535 pp., 2013.

Ji, L., and Peters, A. J.: Assessing vegetation response to drought in the northern Great Plains using vegetation and drought indices, Remote Sens. Environ., 87, 85-98, https://doi.org/10.1016/S0034-4257(03)00174-3, 2003.

Kogan, F., and Guo, W.: Strong 2015–2016 El Niño and implication to global ecosystems from space data, Int J Remote Sens, 38, 161-178, 10.1080/01431161.2016.1259679, 2017.

Liu, W. T., and Juárez, R. I. N.: ENSO drought onset prediction in northeast Brazil using NDVI, Int J Remote Sens, 22, 3483-3501, 10.1080/01431160010006430, 2001.

Lupo, A. R., Garcia, M., Rojas, K., and Gilles, J.: ENSO Related Seasonal Range Prediction over South America, Proceedings, 1, 682, 2017.

Nash, J. E., and Sutcliffe, J. V.: River flow forecasting through conceptual models part I — A discussion of principles, J. Hydrol., 10, 282-290, https://doi.org/10.1016/0022-1694(70)90255-6, 1970.

Porter, J. R., and Semenov, M. A.: Crop responses to climatic variation, Philosophical Transactions of the Royal Society B: Biological Sciences, 360, 2021-2035, 10.1098/rstb.2005.1752, 2005.

Ramirez-Rodrigues, M. A., Asseng, S., Fraisse, C., Stefanova, L., and Eisenkolbi, A.: Tailoring wheat management to ENSO phases for increased wheat production in Paraguay, Climate Risk Management, 3, 24-38, https://doi.org/10.1016/j.crm.2014.06.001, 2014.

Seiler, C., Hutjes, R. W. A., and Kabat, P.: Climate Variability and Trends in Bolivia, J. Appl. Meteorol. Clim., 52, 130-146, 10.1175/Jamc-D-12-0105.1, 2013.

Thompson, L. G., Mosley-Thompson, E., and Arnao, B. M.: El Nino-Southern Oscillation events recorded in the stratigraphy of the tropical Quelccaya ice cap, Peru, Science, 226, 50-53, 10.1126/science.226.4670.50, 1984.

Tippett, M. K., Barnston, A. G., and Li, S.: Performance of Recent Multimodel ENSO Forecasts, J. Appl. Meteorol. Clim., 51, 637-654, 10.1175/jamc-d-11-093.1, 2012.

UN: The Sustainable Development Goals Report 2016, United Nations New York, USA, 2016. UNISDR: Drought Risk Reduction Framework and Practices: Contributing to the Implementation of the Hyogo Framework for Action, United Nations secretariat of the International Strategy for Disaster Reduction (UNISDR), Geneva, Switzerland, 2009.

UNISDR: Making Development Sustainable: The Future of Disaster Risk Management. Global Assessment Report on Disaster Risk Reduction, United Nations Office for Disaster Risk Reduction (UNISDR), Geneva, Switzerland, 266, 2015.

Verbist, K., Amani, A., Mishra, A., and Cisneros, B. J.: Strengthening drought risk management and policy: UNESCO International Hydrological Programme's case studies from Africa and Latin America and the Caribbean, Water Policy, 18, 245-261, 10.2166/wp.2016.223, 2016.

Vicente-Serrano, S. M., Chura, O., López-Moreno, J. I., Azorin-Molina, C., Sanchez-Lorenzo, A., Aguilar, E., Moran-Tejeda, E., Trujillo, F., Martínez, R., and Nieto, J. J.: Spatio-temporal variability of droughts in Bolivia: 1955–2012, Int. J. Climatol., 35, 3024-3040, 10.1002/joc.4190, 2015.

Vuille, M.: Atmospheric circulation over the Bolivian Altiplano during dry and wet periods and extreme phases of the Southern Oscillation, Int. J. Climatol., 19, 1579-1600, 10.1002/(SICI)1097-0088(19991130)19:14<1579::AID-JOC441>3.0.CO;2-N, 1999.

Vuille, M., Bradley, R. S., and Keimig, F.: Interannual climate variability in the Central

Andes and its relation to tropical Pacific and Atlantic forcing, J. Geophys. Res. Atmos., 105, 12447-12460, 10.1029/2000JD900134, 2000.

Wilks, D. S.: Statistical Methods in the Atmospheric Sciences, second ed., Academic Press, 2006.

Please also note the supplement to this comment:
https://www.nat-hazards-earth-syst-sci-discuss.net/nhess-2018-133/nhess-2018-133-AC3-supplement.pdf

**Supplement:**

[Figure]

Fig. R1. Map of mean annual precipitation (July 1981- June 2016) of (a) gauged data* and isohyets** and (b) CHIRPS satellite product. Source: *SENAMHI, **Ministry of Rural Development and Land of Bolivia.

[Figure]

Fig R2. Scatterplot of the day of the year (DOY) and 15-day NDVI from July 1981 to December 2015 for the 23 locations described in Table A1.

[Figure]

Figure R3. Scatterplot of monthly gauged and satellite precipitation data for the 23 studied locations from July 1981 to December 2015.

[Figure]

Fig. R4. RMSE for (a) January and (b) June.

[Figure]

Fig. R5. Map of the Altiplano showing (a) RMSE and (b) ME at the 23 studied locations from July 1981 to June 2016

[Figure]

Fig R6. Correlation coefficient (R2) of the regression of NDVI as the predictand, and precipitation as the predictor for the grids where NDVI better estimate the (a) quinoa and (b) potato yield.

[Figure]

Fig. R7. Comparison of the crop production between the mean crop yield from 1981 to 2016 and the crop yield of 1982/1983 of a) quinoa and b) potato crops.

[Figure]

Fig. R8 NDVI reduction between the NDVI during a strong El Nino 1982-1983 and the mean NDVI from 1981 to 2016 for a) quinoa and b) potato crop.